# Listening to the Brain:
# Multi-Band sEEG Auditory Reconstruction via Dynamic Spatio-Temporal Hypergraphs

**Xueyi Zhang**[1]   **Ruicong Wang**[3]   **Jialu Sun**[1]   **Siqi Cai**[2*]   **Haizhou Li**[1]

[1]School of Artificial Intelligence, The Chinese University of Hong Kong, Shenzhen
[2]School of Intelligence Science and Engineering, Harbin Institute of Technology, Shenzhen
[3] School of Data Science, The Chinese University of Hong Kong, Shenzhen
zhangxueyi@cuhk.edu.cn, ruicongwang@link.cuhk.edu.cn, sunjialu@cuhk.edu.cn
caisiqi@hit.edu.cn, haizhouli@cuhk.edu.cn

## Abstract

Speech is a fundamental form of human communication, and speech perception constitutes the initial stage of language comprehension. Although brain-to-speech interface technologies have made significant progress in recent years, most existing studies focus on neural decoding during speech production. Such approaches heavily rely on articulatory motor regions, rendering them unsuitable for individuals with speech motor impairments, such as those with aphasia or locked-in syndrome. To address this limitation, we construct and release NeuroListen, the first publicly available stereo-electroencephalography (sEEG) dataset specifically designed for auditory reconstruction. It contains over 10 hours of neuralspeech paired recordings from 5 clinical participants, covering a wide range of semantic categories. Building on this dataset, we propose HyperSpeech, a multi-band neural decoding framework that employs dynamic spatio-temporal hypergraph neural networks to capture high-order dependencies across frequency, spatial, and temporal dimensions. Experimental results demonstrate that HyperSpeech significantly outperforms existing methods across multiple objective speech quality metrics, and achieves superior performance in human subjective evaluations, validating its effectiveness and advancement. This study provides a dedicated dataset and modeling framework for auditory speech decoding, offering foundations for neural language processing and assistive communication systems.

## 1   Introduction

Recent advances in braincomputer interfaces (BCIs) have enabled direct speech synthesis from neural signals, offering transformative communication capabilities for individuals with severe speech impairments (1; 2; 3). These technologies promise to restore natural communication ability for patients with conditions such as locked-in syndrome or aphasia (4).

While surface EEG is widely used due to its non-invasive nature, its limited spatial resolution restricts access to deep neural circuits that are critical for speech processing (5; 6). Intracranial EEG (iEEG), including electrocorticography (ECoG) and stereoelectroencephalography (sEEG), offers significantly higher temporal and spatial resolution, making it a powerful tool for decoding neural correlates of speech (6; 7; 8). ECoG-based speech synthesis has demonstrated promising results in recent years (6; 7; 9). In contrast, sEEG provides several unique advantages: it requires only minimally invasive procedures for electrode implantation (10), offers safer long-term monitoring (11; 12), and enables access to both deep and distributed brain regions through spatially sparse but widespread

---

*Corresponding author.

sampling (13; 14; 15; 3). These properties are especially valuable for capturing speech-related dynamics that span bilateral or anatomically distant areas of the brain (16; 17; 18).

Despite these advancements, most existing research has focused on decoding speech production, such as overt or imagined articulation (19; 20; 21; 22). In contrast, auditory speech perception—the neural process of recognizing and understanding speech—constitutes the first stage of language comprehension and communication, yet remains significantly underexplored in neural decoding research. This gap is critical: perception-based decoding not only offers broader clinical applicability (e.g., for non-verbal or locked-in patients), but also provides unique insights into how the brain encodes incoming language information, complementing production-focused approaches.

To address this gap, we introduce NeuroListen, the first publicly available sEEG dataset specifically designed for auditory speech reconstruction. It contains over 10 hours of neuralspeech paired recordings from five clinical participants, covering a diverse set of semantic categories.

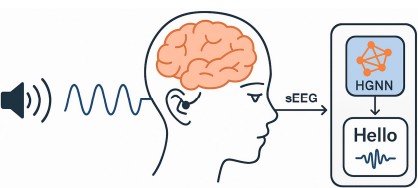

Figure 1: Overview of auditory speech reconstruction from sEEG signals using a hypergraph-based neural decoder.

sEEG enables high-resolution access to both deep and distributed brain regions involved in auditory processing (5; 6). Building on this dataset, we propose HyperSpeech, a novel multi-band decoding framework based on dynamic spatio-temporal hypergraph neural networks, which models complex interactions across frequency, spatial, and temporal dimensions. As shown in Figure 1, the system captures neural responses to heard speech using sEEG and reconstructs intelligible speech through a hypergraph-based decoding pipeline.

Experimental results on the NeuroListen dataset demonstrate that HyperSpeech achieves consistent and significant improvements over multiple competitive baselines. Specifically, it outperforms strong models—including CNN-LSTM (23), Braintalker (24), and FastSpeech (25)—across four objective metrics (PCC: 0.9488, MCD: 1.993, RMSE: 0.2522, STOI: 0.8667) and two human evaluation scores (SMOS: 3.93, CMOS: 4.55). These results highlight its ability to generate intelligible, high-quality speech from sEEG recordings.

## 2  Related Work

**Speech Decoding from Neural Activity: Production and Imagination.** Martin et al. (26) decoded spectro-temporal features of speech from brain activity using ECoG, and Mugler et al. (27) further demonstrated that the full set of American English phonemes can be decoded from ECoG. In (9), Moses et al. explored real-time decoding of perceived and produced speech from high-density ECoG activity during a question-and-answer dialogue task. Angrick et al. (28) explored the use of deep neural networks (3D convolutional neural networks) for reconstructing speech from ECoG recordings. Moses et al. (4) investigated the long-term stability of ECoG recording and its performance in decoding speech over an extensive 81-week recording period in a paralyzed patient with anarthria.

**Neural Reconstruction of Perceived Stimuli: Vision and Audition.** Recent advances in neural decoding have shown impressive progress in reconstructing visual stimuli from brain activity. Zeng et al. (29) proposed CMVDM, which leverages attribute alignment to extract semantics and silhouettes from fMRI, generating high-fidelity images aligned with perceived content. For dynamic visual decoding, Gong et al. (30) introduced NeuroClips, which separates semantic and perceptual pathways for smooth video reconstruction from fMRI. Li et al. (31) demonstrated that even EEG can guide competitive image reconstruction by aligning neural signals with CLIP embeddings via a two-stage diffusion model.

Recent studies have begun to explore auditory reconstruction from non-invasive brain signals. Park et al. and LeBel et al. collected fMRI data when the particiants listened to different auditory stimulus and proposed dataset of fMRI-speech pairs (32; 33). Liu et al. (34) proposed LDM, which used Latent Diffusion Model to reconstruct auditory stimulus from fMRI.

While their work represents a significant advance in auditory reconstruction, the reliance on fMRI comes with inherent limitations such as low temporal resolution and indirect measurement of neural activity. In contrast, our work is the first to reconstruct auditory speech from sEEG, an intracranial

modality with millisecond-level temporal precision and deep-brain access, complements existing fMRI-based approaches.

## 3 NeuroListen Dataset Construction

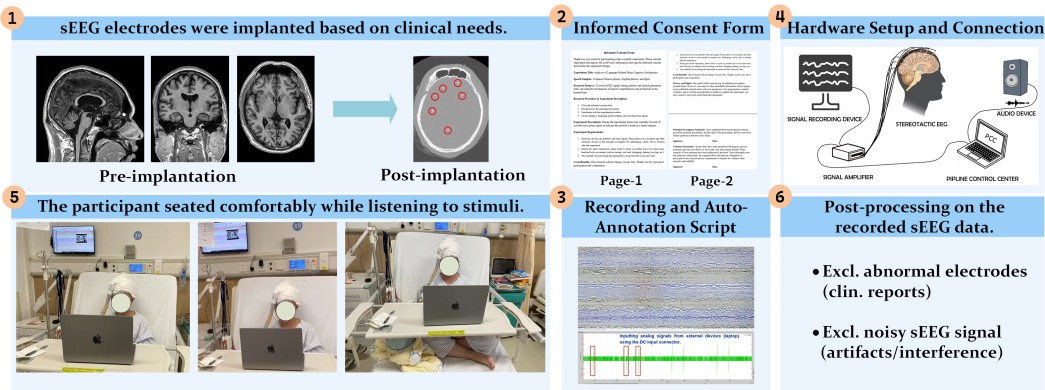

Figure 2: Experimental procedure for sEEG data collection and processing. The workflow includes electrode implantation, informed consent, recording setup, stimulus presentation, data acquisition, and post-processing. Further details are provided in Sections 3 and 4.

### 3.1 Participants

Five patients with epilepsy undergoing neurosurgical treatment were enrolled as the listening subjects in the data collection. They are referred to as the participants. All the subjects were native Mandarin Chinese speaker with basic English conversation skills. The patients ranged in age from 25 to 40 years old, with an average age of 31 years old.

The study was conducted in accordance with the principles embodied in the Declaration of Helsinki. All the patients gave written informed consent to participate in the study. Data collection was conducted under the supervision of experienced doctors to ensure the comfort and safety of the participants. During the recording process, patients were required not to enter any personal identification information. Therefore, this dataset does not contain the identity information of actual users.

### 3.2 Neural Recordings

All the participants were implanted with sEEG electrode shafts to identify epileptogenic foci and all the locations of sEEG electrodes were determined based on each patient's specific epilepsy treatment plan. 8-13 electrode shafts were implanted in each subject. Each shaft contains 8-16 electrode contacts, resulting in a total of 118 - 186 electrode contacts for the subjects. To accurately determine the positions of contacts, we used an open-source MATLAB package LeGUI (35), in which the processing is performed based on Statistical Parametric Mapping toolbox (SPM12) (36). Figure 3 illustrates three views of the depth electrode locations for three participant, where dots of the same color represent electrodes belonging to the same shaft.

### 3.3 Data Acquisition

The participants underwent the implantation of platinum-iridium sEEG electrode shafts (Sinovation (Beijing) Medical Technology SDE-10/12/16, China), featuring a diameter of 0.8 mm and an inter-contact distance of 3.5 mm. Each electrode shaft contained between 10 and 16 electrode contacts. In particular, the placement of all electrodes was determined on the basis of the therapeutic requirements of the patients. sEEG signals were recorded at a sampling rate of 1000 Hz (Nihon Kohden EEG 1200, Tokyo, Japan).

As depicted in Figure 2, a computer was placed in front of the participants, serving as the control center. It delivered the speech stimuli via a speaker. During recording, the computer screen shows a blank screen so as not to distract the participants. All the participants' sEEG signals were recorded.

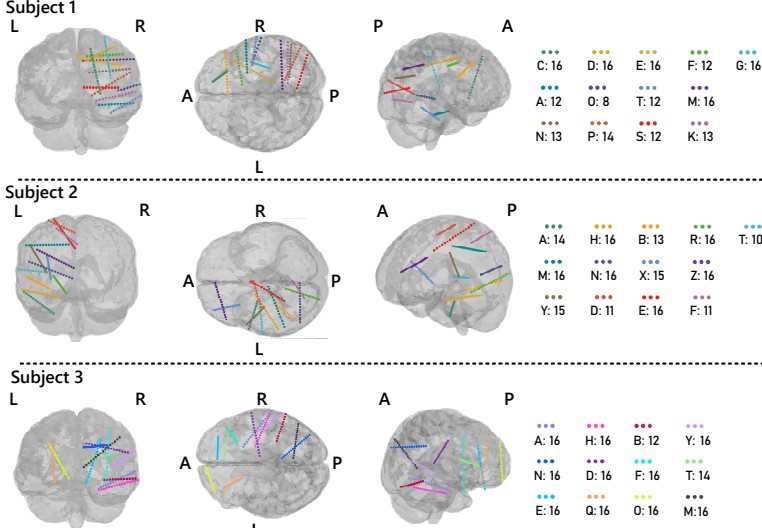

Figure 3: sEEG electrode shaft locations from Subjects 1 to 3. Each dot indicates an individual electrode contact, with contacts of the same color belonging to the same electrode shaft. The trajectories span various cortical and subcortical regions, determined according to each patient's clinical needs. Electrode coordinates were obtained by co-registering pre-implantation magnetic resonance imaging (MRI) with post-implantation computed tomography (CT).

To ensure synchronization between the auditory stimuli and sEEG responses, we employed a Python-scripted tool to play speech stimuli and simultaneously mark the corresponding sEEG responses.

### 3.4 Experiment Protocol

During our experiment, participants were presented with auditory stimuli across three different categories: 44 categories of Mandarin Chinese words, 10 categories of Mandarin Chinese digits (0-9), and 20 categories of English words in each round. The duration designated for listening was set to 2 seconds for each word. At the beginning of each round, each participant was given a 5-second interval to get ready, where a prompt "Please listen to the speech attentively" is played, that is followed by a "ding" sound to represent the start of the attended speech content.

To avoid fatigue, the participants took a 5 to 10-minute break between two rounds. Additionally, several familiarization trials were conducted to ensure that the subjects understood the experimental procedures before recording. As a result, the NeuroListen dataset comprises more than 10 hours of neuralspeech paired recordings in total.

### 3.5 sEEG Alignment and Annotation

In this study, sEEG and speech signals were aligned via a designed experimental setup and annotation process. During data collection, the computer program sent a marker signal to the DC electrode and played speech simultaneously as shown in Figure 2, ensuring the speech and sEEG onset were aligned. For annotation, the speech segmentation start point was determined by considering participant task performance variations and software recording lags. The starting point was selected with a precision of 0.01 - 0.005 seconds, enabling accurate signal alignment for speech neural analysis.

## 4 Data Preprocessing

### 4.1 Data Loading

The NeuroListen dataset is publicly available for research use (`https://zenodo.org/records/17426506`). To simplify the use of the data, we have preprocessed the sEEG signals and corresponding

speech signals. Specifically, files with the extension _seeg.npy contain the processed sEEG data for each participant, while files ending in _mel.npy contain the corresponding mel-spectrogram of the speech.

## 4.2  Neural Signal Preprocessing

First, we excluded electrodes identified in epileptologists' reports as showing abnormal epileptiform discharges (37). Electrodes marked as epileptogenic or exhibiting frequent spikes, sharp-slow waves, or high-frequency discharges (as documented in clinical reports) were excluded from analysis. Specifically, 61, 27, 46, 22, 37 electrodes were removed from the patients respectively.

Subsequently, bipolar referencing was applied to the remaining sEEG signals (38). Previous studies have highlighted the critical role of high-gamma frequency (HGA) and low-frequency signal (LFS) features in synthesizing speech from brain signals (8; 39; 6). Accordingly, we followed the preprocessing methods used in previous research to extract the LFS and HGA frequency bands (6). Additionally, we tested broadband signals (BBS), which combine both LFS and HGA sEEG features, to provide a comprehensive perspective and evaluate their combined contributions to speech synthesis performance. Specifically, to compute HGA, we first band-passed the signals in the high-gamma frequency range ($70 - 150$ Hz), then calculated the analytic amplitude of these signals, and finally downsampled them to 200 Hz. For LFS, we applied a low-pass anti-aliasing filter with a cutoff frequency of 100 Hz before downsampling the signals to 200 Hz. Lastly, we normalized the extracted HGA and LFS signals from each sEEG electrode within each 2-second window. The same neural signal preprocessing pipeline was applied consistently across all baseline and proposed models to ensure fair comparison.

## 4.3  Speech Signal Preprocessing

We used LibROSA, a commonly adopted Python library for speech processing (40), to downsample the speech signals to 16 kHz and extract the mel-spectrograms. To capture the temporal dynamics of the specch signal, a window length of 64 milliseconds and a hop length of 20 milliseconds were set. Additionally, we set the number of bins in the mel-spectrogram to 80, aiming to capture sufficiently detailed frequency information to describe the speech signals (41).

# 5  Methods

Hypergraphs generalize traditional graphs by enabling edges to connect multiple nodes, making them well-suited for modeling high-order relationships. They have been successfully applied in diverse domains such as action recognition, time-series analysis (42; 43; 44), demonstrating their effectiveness in capturing complex spatial and temporal dependencies. Inspired by these advances, we adopt HGNNs to model spatio-temporal dynamics in sEEG signals for improved auditory reconstruction.

Figure 4l presents the overall architecture of HyperSpeech, which performs spatio-temporal feature extraction on dual-band sEEG signals via dynamic hypergraphs. Raw signals are first fused within each electrode shaft and passed through convolutional layers to extract spatial features, then segmented into temporal windows. At each time step, a spatial hypergraph captures inter-regional spatial dependencies, while a temporal hypergraph models temporal relations. After spatio-temporal hypergraph convolutions, multi-band features are fused and processed by a Bi-LSTM to extract fine-grained temporal dynamics. The model then predicts mel-spectrograms, which are converted into high-quality speech using a HiFi-GAN decoder.

## 5.1  Spatial Hypergraph Construction

The raw sEEG signal is denoted as $X \in \mathbb{R}^{C \times d}$, where $C$ is the total number of channels and $d$ is the number of time steps. The data from each of the $N$ electrode shafts is represented as $x_n \in \mathbb{R}^{C_n \times d}$, where $C_n$ denotes the number of channels on shaft $n$, with $n = 1, 2, \ldots, N$. To integrate information across channels within each shaft, we apply a fully connected layer for intra-shaft channel fusion.

$$X_1 = \text{concat}(FC_1(x_1), FC_2(x_2), ..., FC_N(x_N)). \tag{1}$$

Here, $FC_n$ denotes the fully connected fusion layer for the $n$-th electrode shaft. We then apply 1D convolution to extract spatio-temporal features from the fused sEEG signals.

$$X_2 = \text{Conv1d}(X_1). \tag{2}$$

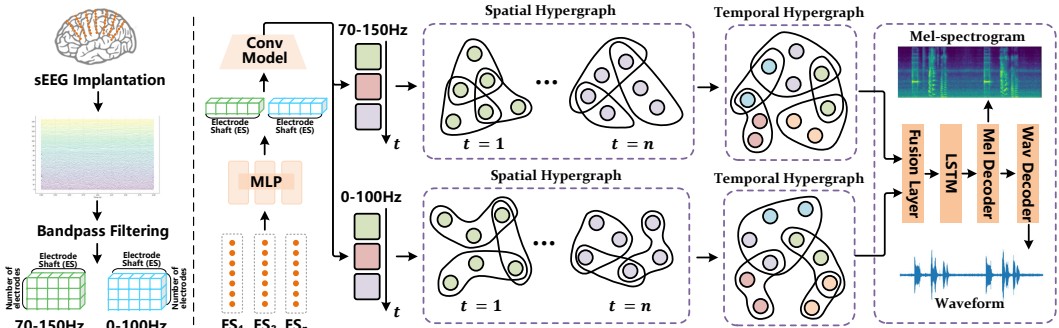

Figure 4: An overview of our proposed HyperSpeech. The model utilizes spatiotemporal hypergraphs to extract spatiotemporal features from two frequency bands of sEEG signals, effectively capturing higher-order spatiotemporal dependencies to enable accurate sEEG signal decoding for speech.

Given the notable temporal evolution and structural variability of sEEG signals, we divide the data into $T$ temporal windows, represented as $X_3 \in \mathbb{R}^{T \times N \times d_1}$. A spatial hypergraph $\mathcal{G}^t_{spatial} = \{\mathcal{V}^t_{spatial}, \mathcal{E}^t_{spatial}\}$ is constructed within each window to capture the dynamic topological patterns over time. At each time step $t$, the spatial nodes $\mathcal{V}^t_{\text{spatial}}$ represent spatial features of different brain regions, and the hyperedges $\mathcal{E}^t_{\text{spatial}}$ capture high-order inter-regional relationships. For each node, we compute the Euclidean distances to all other nodes and select its $k$ nearest neighbors. By connecting each node to its $k$ neighbors, a hyperedge is formed, effectively modeling spatial dependencies across brain regions. The spatial hypergraph can be represented in the form of a neighborhood matrix.

$$(H^t_s)_{i,j} = \begin{cases} 1, & \text{node } i \text{ is included in hyperedge } j, \\ 0, & \text{otherwise.} \end{cases} \tag{3}$$

Based on the constructed neighborhood matrix $(H^t_s)_{i,j}$, the model applies hypergraph convolution to capture high-order relationships among spatial nodes. The spatial hypergraph convolution at time step $t$ can be formulated as:

$$(X^t_{spatial})^{(l+1)} = \sigma((D^t_s)_v^{-\frac{1}{2}}(H_s)^T W^t_s (D^t_s)_e^{-\frac{1}{2}} H^t_s (D^t_s)_v^{-\frac{1}{2}} (X^t_{spatial})^{(l)} (\Theta^t_s)^{(l)}). \tag{4}$$

Here, $\sigma$ denotes the activation function, and $(X^t_{\text{spatial}})^{(l)}$ represents the node features at layer $l$, with $(X^t_{\text{spatial}})^{(1)} = X^t_3$. $(D_s)_v$ and $(D^t_s)_e$ denote the degree matrices of nodes and hyperedges, respectively, while $W^t_s$ is the hyperedge weight matrix. $(\Theta^t_s)^{(l)}$ represents the learnable parameters of the $l$-th layer. Subsequently, global average pooling is applied along the spatial dimension to aggregate spatial features at each time step, yielding $X^t_4$. These temporal representations $X^t_4$ are then fed into the temporal hypergraph to further extract high-order temporal dependencies.

## 5.2 Temporal Hypergraph Modeling

After extracting spatial features from sEEG signals using spatial hypergraphs, we construct a temporal hypergraph to further capture high-order temporal dependencies. The temporal hypergraph is defined as $\mathcal{G}_{\text{temporal}} = (\mathcal{V}_{\text{temporal}}, \mathcal{E}_{\text{temporal}})$, where each node $v \in \mathcal{V}_{\text{temporal}}$ represents the spatial feature at a specific time step. The hyperedges $\mathcal{E}_{\text{temporal}}$ are constructed by computing the Euclidean distance between each time frame and all others, connecting each time step to its $k$ most similar neighbors. In this way, the temporal hyperedges can capture high-order dependencies across time. The temporal hypergraph convolution is formulated as:

$$X^{(l+1)}_{temporal} = \sigma \left( D_v^{-\frac{1}{2}} H^T W D_e^{-\frac{1}{2}} H D_v^{-\frac{1}{2}} X^{(l)}_{temporal} \Theta^{(l)} \right). \tag{5}$$

Here, $X^{(1)}_{\text{temporal}} = X_4$. After two parallel spatio-temporal hypergraph convolutions, we obtain the spatio-temporal representations for the two frequency bands, denoted as $X^{(\text{HGA})}_5$ and $X^{(\text{LFS})}_5$. A fully connected layer is then applied as a fusion layer to integrate these multi-band features.

$$X_6 = FC(X^{(HGA)}_5, X^{(LFS)}_5). \tag{6}$$

This yields the multi-band spatio-temporal feature representation $X_6 \in \mathbb{R}^{T \times d_5}$. To further capture fine-grained temporal dependencies, $X_6$ is fed into a bidirectional long short-term memory network (Bi-LSTM) for temporal modeling.

$$X_7 = BiLSTM(X_6). \tag{7}$$

Finally, we obtain the spatio-temporally encoded sEEG representation $X_7$, which is fed into the speech decoder to predict mel-spectrograms and generate the corresponding speech waveform.

### 5.3 Mel-Spectrogram and Speech Synthesis

In the decoding stage, the spatio-temporal representation $X_7$ is projected via a fully connected layer to obtain the mel-spectrogram features $X_{\text{mel}}$. To convert the mel-spectrogram into high-quality audio, we adopt HiFi-GAN (45), a vocoder based on a generative adversarial network (GAN) architecture. It consists of a generator and two discriminators (multi-scale and multi-period), enabling efficient and realistic waveform synthesis from mel-spectrograms through adversarial training. Finally, the predicted $X_{\text{mel}}$ is fed into the HiFi-GAN decoder to produce the corresponding speech waveform $X_{\text{wave}}$. This decoding process ensures high-quality speech synthesis suitable for practical neural speech reconstruction tasks.

The inference pipeline of HyperSpeech is presented in Algorithm 1, detailing the sequential computation from multi-band sEEG input to waveform reconstruction. Each stage corresponds to a core functional block in the proposed framework and is aligned with the equations defined in our main paper.

---

**Algorithm 1** HyperSpeech Inference Pipeline

---

**Require:** Raw sEEG signals $X^{(f)} \in \mathbb{R}^{C \times d}$ for frequency band $f \in \{\text{HGA}, \text{LFS}\}$, number of electrode shafts $N$, number of time windows $T$, number of neighbors $K$, number of convolution layers $L$

1: **for** $f \in \{\text{HGA}, \text{LFS}\}$ **do**
2:      $X_1^{(f)} \leftarrow$ intra-shaft channel fusion via Main Eq. (1)
3:      $X_2^{(f)} \leftarrow$ spatio-temporal feature extraction via Main Eq. (2)
4:      $X_3^{(f)} \leftarrow$ windowed representation
5:      **for** $t = 1$ to $T$ **do**
6:          $\mathcal{G}_{spatial}^t \leftarrow$ spatial hypergraph construction via Main Eq. (3)
7:          $X_{spatial}^{t(l)} \leftarrow$ spatial hypergraph convolution via Main Eq. (4)
8:      **end for**
9:      $\mathcal{G}_{temporal} \leftarrow$ temporal hypergraph construction
10:    $X_5^{(f)} \leftarrow$ temporal hypergraph convolution via Main Eq. (5)
11: **end for**
12: $X_6 \leftarrow$ multi-band feature fusion via Main Eq. (6)
13: $X_7 \leftarrow$ Bi-LSTM modeling via Main Eq. (7)
14: $X_{\text{mel}} \leftarrow$ mel-spectrogram projection
15: $X_{\text{wave}} \leftarrow$ waveform generation (HiFi-GAN)
**Ensure:** Reconstructed speech waveform $X_{\text{wave}}$

---

## 6 Experiments and Results

### 6.1 Implement Details

Our method was implemented using PyTorch 1.11.0 with CUDA 11.3. All models were trained for 100 epochs using the Adam optimizer with a batch size of 16. The initial learning rate was set to $3 \times 10^{-4}$ and decayed to $5 \times 10^{-6}$ following a cosine annealing schedule. For each subject, we performed 5-fold cross-validation, using an 80% and 20% split for training and testing in each fold.

All experiments were conducted on an NVIDIA RTX 4090 GPU. Our objective evaluation metrics include PCC (24; 46; 47), MCD (24; 47), RMSE (24), and STOI (47), covering aspects such as correlation, spectral distortion, and speech intelligibility. In addition, the Mean Opinion Score (MOS)

(25) is employed as the primary evaluation metric to assess the performance of the generated audio in terms of similarity (SMOS) and clarity (CMOS) (25), with comprehensive definitions provided in the appendix. The subjective evaluations (SMOS and CMOS) were collected from 30 independent raters to ensure unbiased assessment of the reconstructed speech quality.

## 6.2 Baseline Methods

In this study, we compare our method against three representative baseline models: CNN-LSTM (23), Braintalker (24), and FastSpeech (25). The CNN-LSTM model combines convolutional neural networks (CNNs) for spatial feature extraction with long short-term memory (LSTM) networks for modeling temporal dependencies, and is widely used for EEG sequence modeling. Braintalker adopts an end-to-end neural architecture and leverages self-supervised learning for EEG feature representation. FastSpeech, a widely adopted baseline in speech synthesis, is built upon the Transformer architecture and is known for its robustness and synthesis quality.

## 6.3 Main Results

Table 1: Comparison of HyperSpeech with other state-of-the-art methods across different subjects.

| Subjects | Methods | PCC↑ | MCD↓ | RMSE↓ | STOI↑ | SMOS↑ | CMOS↑ |
|---|---|---|---|---|---|---|---|
| Subject1 | CNN-LSTM (23) | 0.9147 | 2.469 | 0.3346 | 0.8209 | $3.50_{\pm 0.26}$ | $4.03_{\pm 0.13}$ |
| | Braintalker (24) | 0.8927 | 2.508 | 0.3661 | 0.8280 | $3.23_{\pm 0.17}$ | $3.80_{\pm 0.16}$ |
| | FastSpeech (25) | 0.8806 | 2.391 | 0.3713 | 0.8118 | $3.27_{\pm 0.25}$ | $4.13_{\pm 0.11}$ |
| | Ours | 0.9368 | 2.286 | 0.3285 | 0.8454 | $3.63_{\pm 0.19}$ | $4.33_{\pm 0.13}$ |
| Subject2 | CNN-LSTM (23) | 0.9086 | 2.450 | 0.3528 | 0.7887 | $3.60_{\pm 0.22}$ | $4.13_{\pm 0.24}$ |
| | Braintalker (24) | 0.9024 | 2.474 | 0.3590 | 0.8342 | $3.27_{\pm 0.24}$ | $3.87_{\pm 0.26}$ |
| | FastSpeech (25) | 0.9192 | 2.530 | 0.3230 | 0.8099 | $3.40_{\pm 0.28}$ | $4.23_{\pm 0.19}$ |
| | Ours | 0.9125 | 2.506 | 0.3005 | 0.8439 | $3.83_{\pm 0.14}$ | $4.43_{\pm 0.12}$ |
| Subject3 | CNN-LSTM (23) | 0.9421 | 1.688 | 0.2453 | 0.8758 | $3.63_{\pm 0.32}$ | $4.53_{\pm 0.26}$ |
| | Braintalker (24) | 0.9461 | 1.805 | 0.2480 | 0.8659 | $3.53_{\pm 0.36}$ | $4.63_{\pm 0.21}$ |
| | FastSpeech (25) | 0.9402 | 1.877 | 0.2551 | 0.8739 | $3.77_{\pm 0.21}$ | $4.73_{\pm 0.26}$ |
| | Ours | 0.9523 | 1.716 | 0.2463 | 0.8550 | $3.90_{\pm 0.18}$ | $4.60_{\pm 0.27}$ |
| Subject4 | CNN-LSTM (23) | 0.9767 | 1.860 | 0.1821 | 0.8851 | $4.03_{\pm 0.12}$ | $4.63_{\pm 0.17}$ |
| | Braintalker (24) | 0.9660 | 1.688 | 0.2080 | 0.8643 | $3.80_{\pm 0.32}$ | $4.47_{\pm 0.12}$ |
| | FastSpeech (25) | 0.9508 | 1.704 | 0.2301 | 0.8842 | $3.53_{\pm 0.29}$ | $4.43_{\pm 0.33}$ |
| | Ours | 0.9781 | 1.706 | 0.1787 | 0.8955 | $4.13_{\pm 0.24}$ | $4.73_{\pm 0.11}$ |
| Subject5 | CNN-LSTM (23) | 0.9548 | 1.895 | 0.2226 | 0.8806 | $4.03_{\pm 0.26}$ | $4.33_{\pm 0.16}$ |
| | Braintalker (24) | 0.9551 | 1.820 | 0.2270 | 0.8758 | $4.10_{\pm 0.20}$ | $4.47_{\pm 0.19}$ |
| | FastSpeech (25) | 0.9573 | 1.922 | 0.2190 | 0.8811 | $3.83_{\pm 0.31}$ | $4.63_{\pm 0.22}$ |
| | Ours | 0.9645 | 1.750 | 0.2067 | 0.8936 | $4.13_{\pm 0.17}$ | $4.67_{\pm 0.10}$ |
| Average | CNN-LSTM (23) | 0.9393 | 2.072 | 0.2675 | 0.8502 | $3.76_{\pm 0.24}$ | $4.33_{\pm 0.19}$ |
| | Braintalker (24) | 0.9325 | 2.059 | 0.2816 | 0.8536 | $3.59_{\pm 0.26}$ | $4.25_{\pm 0.19}$ |
| | FastSpeech (25) | 0.9296 | 2.085 | 0.2797 | 0.8521 | $3.56_{\pm 0.27}$ | $4.43_{\pm 0.22}$ |
| | Ours | 0.9488 | 1.993 | 0.2522 | 0.8667 | $3.92_{\pm 0.18}$ | $4.55_{\pm 0.15}$ |

Through cross-subject evaluation, the results demonstrate that HyperSpeech consistently outperforms baseline models across all four objective metrics, validating its effectiveness in modeling spatio-temporal features from sEEG signals.

Specifically, HyperSpeech achieved strong performance in PCC across all participants, with a notable value of 0.9368 on Subject 1—substantially higher than baseline methods. In terms of spectral fidelity, the model obtained an MCD of 2.286, indicating reduced spectral distortion. Furthermore, it also achieved competitive results in RMSE and STOI, reaching 0.3285 and 0.8454 respectively, reflecting superior error control and intelligibility preservation.

Further subject-wise analysis reveals that HyperSpeech consistently achieved top performance across individuals. On Subject 4 and Subject 5, the model yielded PCC scores of 0.9781 and 0.9645, and STOI scores of 0.8955 and 0.8936, respectively—all substantially outperforming competing baselines. These results highlight the model's capability to capture complex neural spatio-temporal dependencies while maintaining strong generalization across subjects.

Moreover, HyperSpeech demonstrated superior subjective quality with an average SMOS of 3.92 and CMOS of 4.55 across all subjects, outperforming all baseline models. Similarity MOS (SMOS) measures the perceived similarity between the reconstructed and reference speech, while Clarity

MOS (CMOS) evaluates the clarity and intelligibility of the speech. Both metrics are rated on a scale from 1 to 5, with higher scores indicating better quality.

## 6.4 Ablation Study

**Effectiveness of Time and Space Hypergraphs in Our Model.** We evaluated the contribution of the spatial and temporal hypergraph modules to the overall model performance, as summarized in Tables 2. When the spatial hypergraph module was removed, the PCC dropped by 0.0136 and the STOI decreased by 0.0129. Removing the temporal hypergraph module resulted in a larger performance degradation, with a 0.0246 drop in PCC and a 0.0187 drop in STOI. These results indicate that both spatial and temporal hypergraphs play critical roles in modeling high-order spatio-temporal dependencies and inter-regional relationships in sEEG signals. In particular, the spatial hypergraph significantly enhances the model's ability to capture complex spatial structures, while the temporal hypergraph effectively improves the modeling of temporal dependencies.

Table 2: Effectiveness of each part of our HyperSpeech.

| Model | PCC | MCD | RMSE | STOI |
|---|---|---|---|---|
| w/o SHG | 0.9352 | 2.055 | 0.2664 | 0.8538 |
| w/o THG | 0.9242 | 2.068 | 0.2732 | 0.8480 |
| **Ours** | **0.9488** | **1.993** | **0.2522** | **0.8667** |

**Effectiveness of Different Frequency Bands on Model Performance.** We further investigated the impact of different frequency bands on spatio-temporal hypergraph modeling, as shown in Tables 3. When using only the LFS band, the model achieved a PCC of 0.9312 and a STOI of 0.8546. With only the HGA band, performance improved to a PCC of 0.9376 and a STOI of 0.8523. In comparison, the proposed model achieved the best performance when both frequency bands were fused, reaching a PCC of 0.9488 and a STOI of 0.8667.

These results indicate that the LFS and HGA bands contribute differently to model performance, and their complementary properties play a vital role in auditory decoding. The integration of both frequency bands enables the model to capture a broader range of spatio-temporal dynamics in the sEEG signals, thereby significantly enhancing decoding accuracy.

Table 3: Quantitative Evaluation of Model Performance Across Different Frequency Bands.

| sEEG feature | PCC | MCD | RMSE | STOI |
|---|---|---|---|---|
| LFS | 0.9312 | 2.072 | 0.2694 | 0.8546 |
| HGA | 0.9376 | 2.037 | 0.2611 | 0.8523 |
| **BBS(Ours)** | **0.9488** | **1.993** | **0.2522** | **0.8667** |

**Comparison between HyperSpeech and GNN Models.** To further validate the effectiveness of hypergraph structures in spatio-temporal feature extraction, we compared HyperSpeech with traditional graph neural networks (GNNs). As shown in Tables 4, HyperSpeech achieved consistent improvements across all four evaluation metrics: PCC increased by 0.0103, STOI improved by 0.0133, while MCD and RMSE decreased by 0.076 and 0.0121, respectively.

These results highlight the advantage of hypergraph-based modeling in capturing high-order spatio-temporal dependencies from sEEG signals. Unlike conventional GNNs that rely on pairwise connections, hypergraphs can represent multi-way relationships among nodes, which is particularly beneficial for modeling coordinated neural activity across multiple brain regions. This ability to encode higher-order interactions allows HyperSpeech to more effectively capture the intrinsic structure of brain dynamics, leading to superior performance in neural speech decoding tasks.

Table 4: Performance Comparison between HyperSpeech model and GNN model

| Model | PCC | MCD | RMSE | STOI |
|---|---|---|---|---|
| GNN | 0.9385 | 2.069 | 0.2643 | 0.8534 |
| **HyperSpeech(Ours)** | **0.9488** | **1.993** | **0.2522** | **0.8667** |

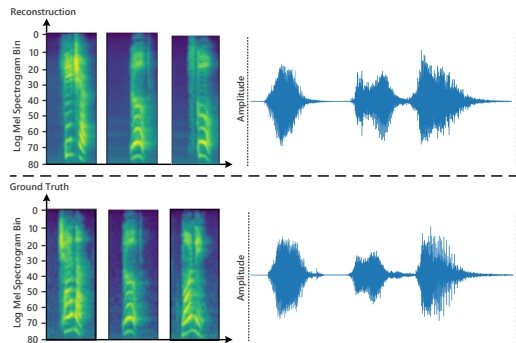

Figure 5: Qualitative Comparison Between Reconstructed and Ground-Truth Speech

**Case Study: Visualization of Reconstruction Quality.** To qualitatively assess the performance of our proposed model, we present a case-wise comparison between reconstructed speech and ground-truth audio in both the mel-spectrogram and waveform domains, as shown in Figure 5. The top row shows the outputs generated by HyperSpeech, while the bottom row presents the corresponding ground-truth references.

As observed in the mel-spectrograms, the reconstructed samples exhibit clear harmonic structures and formant contours that closely resemble those in the ground-truth signals. Key speech dynamics, such as formant transitions and energy distributions, are well preserved, indicating the model's ability to capture fine-grained acoustic features from neural inputs.

In the waveform comparison, the generated speech signals demonstrate smooth amplitude contours and natural prosodic patterns, with high perceptual similarity to the ground-truth audio. These visual results further support our quantitative findings, confirming that HyperSpeech can generate intelligible and high-fidelity speech from sEEG recordings.

# 7 Conclusion

In this work, we introduced NeuroListen, the first publicly available sEEG dataset tailored for auditory speech reconstruction, addressing a long-standing gap in auditory perception-based neural decoding. Leveraging this dataset, we proposed HyperSpeech, a novel multi-band decoding framework based on dynamic spatio-temporal hypergraph neural networks. By capturing high-order dependencies across spatial, temporal, and frequency dimensions, our model enables accurate and intelligible reconstruction of perceived speech from intracranial neural recordings. Comprehensive experiments across five clinical participants demonstrate that HyperSpeech consistently outperforms strong baselines—including CNN-LSTM, Braintalker, and FastSpeech—on both objective metrics and subjective evaluations. These contributions advance the frontier of neural speech decoding and lay a solid foundation for future research into neural language processing and assistive communication technologies.

# 8 Limitations & Future Work

A major limitation is the variability in electrode placement across subjects, which limits cross-subject generalization. Future work will develop transferable decoding frameworks and investigate hierarchical speech representations across cortical regions, as well as the neural mechanisms underlying multilingual speech processing and cross-language generalization.

# 9 Acknowledgements

This study was funded by Shenzhen Science and Technology Program (Shenzhen Key Laboratory, Grant No. ZDSYS20230626091302006), Shenzhen Science and Technology Research Fund (Fundamental Research Key Project, Grant No. JCYJ20220818103001002), and Program for Guangdong Introducing Innovative and Entrepreneurial Teams (Grant No. 2023ZT10X044).

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

# NeurIPS Paper Checklist

1. **Claims**

   Question: Do the main claims made in the abstract and introduction accurately reflect the paper's contributions and scope?

   Answer: [Yes]

   Justification: The paper clearly states and consistently supports its primary contributions in both the abstract and introduction. Specifically:

   - It introduces **NeuroListen**, the *first publicly available sEEG dataset* for auditory speech reconstruction, addressing a key gap in auditory perception-based neural decoding.
   - It proposes **HyperSpeech**, a novel *multi-band decoding framework* based on *spatio-temporal hypergraph neural networks (HGNNs)*, capable of modeling high-order dependencies in sEEG signals.
   - It claims *significant improvements* over strong baselines (CNN-LSTM, Braintalker, FastSpeech), supported by both objective metrics (PCC, MCD, RMSE, STOI) and human evaluations (CMOS, SMOS).
   - These claims are empirically validated across **five clinical subjects**, demonstrating both the *effectiveness and generalizability* of the proposed method.

   Thus, the abstract and introduction faithfully reflect the scope and contributions substantiated by the theoretical framework and experimental results.

   Guidelines:

   - The answer NA means that the abstract and introduction do not include the claims made in the paper.
   - The abstract and/or introduction should clearly state the claims made, including the contributions made in the paper and important assumptions and limitations. A No or NA answer to this question will not be perceived well by the reviewers.
   - The claims made should match theoretical and experimental results, and reflect how much the results can be expected to generalize to other settings.
   - It is fine to include aspirational goals as motivation as long as it is clear that these goals are not attained by the paper.

2. **Limitations**

   Question: Does the paper discuss the limitations of the work performed by the authors?

   Answer: [Yes]

   Justification: We explicitly discuss the limitations in the "Limitations and Future Work" section of the paper. A key limitation lies in the variability of electrode shaft configurations across participants, which introduces challenges in cross-subject generalization. While our model demonstrates strong within-subject performance, extending it to robust, subject-independent decoding remains non-trivial due to differences in electrode placement and signal distribution. Addressing these challenges—particularly through transferable model design, broader validation, and scalable computation—will be the focus of our future work.

   Guidelines:

   - The answer NA means that the paper has no limitation while the answer No means that the paper has limitations, but those are not discussed in the paper.
   - The authors are encouraged to create a separate "Limitations" section in their paper.
   - The paper should point out any strong assumptions and how robust the results are to violations of these assumptions (e.g., independence assumptions, noiseless settings, model well-specification, asymptotic approximations only holding locally). The authors should reflect on how these assumptions might be violated in practice and what the implications would be.
   - The authors should reflect on the scope of the claims made, e.g., if the approach was only tested on a few datasets or with a few runs. In general, empirical results often depend on implicit assumptions, which should be articulated.

- The authors should reflect on the factors that influence the performance of the approach. For example, a facial recognition algorithm may perform poorly when image resolution is low or images are taken in low lighting. Or a speech-to-text system might not be used reliably to provide closed captions for online lectures because it fails to handle technical jargon.
- The authors should discuss the computational efficiency of the proposed algorithms and how they scale with dataset size.
- If applicable, the authors should discuss possible limitations of their approach to address problems of privacy and fairness.
- While the authors might fear that complete honesty about limitations might be used by reviewers as grounds for rejection, a worse outcome might be that reviewers discover limitations that aren't acknowledged in the paper. The authors should use their best judgment and recognize that individual actions in favor of transparency play an important role in developing norms that preserve the integrity of the community. Reviewers will be specifically instructed to not penalize honesty concerning limitations.

3. **Theory assumptions and proofs**

   Question: For each theoretical result, does the paper provide the full set of assumptions and a complete (and correct) proof?

   Answer: [Yes]

   Justification: We provide a theoretical justification of the proposed spatio-temporal hypergraph modeling in the Appendix. Specifically, we analyze the representation capacity of dynamic hypergraphs for modeling high-order spatio-temporal dependencies in sEEG signals. The assumptions and mathematical formulations underlying the hypergraph construction and convolution are clearly stated, and a formal proof sketch is included to support the design of our HyperSpeech framework.

   Guidelines:
   - The answer NA means that the paper does not include theoretical results.
   - All the theorems, formulas, and proofs in the paper should be numbered and cross-referenced.
   - All assumptions should be clearly stated or referenced in the statement of any theorems.
   - The proofs can either appear in the main paper or the supplemental material, but if they appear in the supplemental material, the authors are encouraged to provide a short proof sketch to provide intuition.
   - Inversely, any informal proof provided in the core of the paper should be complemented by formal proofs provided in appendix or supplemental material.
   - Theorems and Lemmas that the proof relies upon should be properly referenced.

4. **Experimental result reproducibility**

   Question: Does the paper fully disclose all the information needed to reproduce the main experimental results of the paper to the extent that it affects the main claims and/or conclusions of the paper (regardless of whether the code and data are provided or not)?

   Answer: [Yes]

   Justification: We provide detailed descriptions of all aspects necessary to reproduce our experimental results. This includes the structure of our proposed HyperSpeech model, the training configuration , and the evaluation metrics. We also report all key hyperparameters (e.g., optimizer, learning rate, batch size, epoch number) and describe the data preprocessing and alignment pipeline used on the NeuroListen dataset. All experiments were conducted on a standardized hardware platform (NVIDIA RTX 4090), and training/testing splits were consistent across all compared methods. Further implementation details and dataset usage instructions are included in the supplementary material to facilitate reproducibility.

   Guidelines:
   - The answer NA means that the paper does not include experiments.
   - If the paper includes experiments, a No answer to this question will not be perceived well by the reviewers: Making the paper reproducible is important, regardless of whether the code and data are provided or not.

- If the contribution is a dataset and/or model, the authors should describe the steps taken to make their results reproducible or verifiable.
- Depending on the contribution, reproducibility can be accomplished in various ways. For example, if the contribution is a novel architecture, describing the architecture fully might suffice, or if the contribution is a specific model and empirical evaluation, it may be necessary to either make it possible for others to replicate the model with the same dataset, or provide access to the model. In general. releasing code and data is often one good way to accomplish this, but reproducibility can also be provided via detailed instructions for how to replicate the results, access to a hosted model (e.g., in the case of a large language model), releasing of a model checkpoint, or other means that are appropriate to the research performed.
- While NeurIPS does not require releasing code, the conference does require all submissions to provide some reasonable avenue for reproducibility, which may depend on the nature of the contribution. For example
  (a) If the contribution is primarily a new algorithm, the paper should make it clear how to reproduce that algorithm.
  (b) If the contribution is primarily a new model architecture, the paper should describe the architecture clearly and fully.
  (c) If the contribution is a new model (e.g., a large language model), then there should either be a way to access this model for reproducing the results or a way to reproduce the model (e.g., with an open-source dataset or instructions for how to construct the dataset).
  (d) We recognize that reproducibility may be tricky in some cases, in which case authors are welcome to describe the particular way they provide for reproducibility. In the case of closed-source models, it may be that access to the model is limited in some way (e.g., to registered users), but it should be possible for other researchers to have some path to reproducing or verifying the results.

5. **Open access to data and code**

   Question: Does the paper provide open access to the data and code, with sufficient instructions to faithfully reproduce the main experimental results, as described in supplemental material?

   Answer: [Yes]

   Justification: We provide the download links and access instructions in the main paper (Section 4.1). And main paper include detailed guidance, covering data preprocessing, model training, and evaluation procedures.

   Guidelines:

   - The answer NA means that paper does not include experiments requiring code.
   - Please see the NeurIPS code and data submission guidelines (`https://nips.cc/public/guides/CodeSubmissionPolicy`) for more details.
   - While we encourage the release of code and data, we understand that this might not be possible, so "No" is an acceptable answer. Papers cannot be rejected simply for not including code, unless this is central to the contribution (e.g., for a new open-source benchmark).
   - The instructions should contain the exact command and environment needed to run to reproduce the results. See the NeurIPS code and data submission guidelines (`https://nips.cc/public/guides/CodeSubmissionPolicy`) for more details.
   - The authors should provide instructions on data access and preparation, including how to access the raw data, preprocessed data, intermediate data, and generated data, etc.
   - The authors should provide scripts to reproduce all experimental results for the new proposed method and baselines. If only a subset of experiments are reproducible, they should state which ones are omitted from the script and why.
   - At submission time, to preserve anonymity, the authors should release anonymized versions (if applicable).
   - Providing as much information as possible in supplemental material (appended to the paper) is recommended, but including URLs to data and code is permitted.

6. **Experimental setting/details**

   Question: Does the paper specify all the training and test details (e.g., data splits, hyper-parameters, how they were chosen, type of optimizer, etc.) necessary to understand the results?

   Answer: [Yes]

   Justification: The paper provides comprehensive experimental settings necessary to reproduce and understand the results. We describe all key hyperparameters, including the optimizer (Adam), learning rate schedule (cosine annealing from $3\times10^{-4}$ to $5\times10^{-6}$), batch size (16), and number of training epochs (100). Training and evaluation were conducted on an NVIDIA RTX 4090 GPU. Details regarding data splits, signal preprocessing, and model architecture are included in Section 3-5, with additional implementation information provided in the supplementary material.

   Guidelines:

   - The answer NA means that the paper does not include experiments.
   - The experimental setting should be presented in the core of the paper to a level of detail that is necessary to appreciate the results and make sense of them.
   - The full details can be provided either with the code, in appendix, or as supplemental material.

7. **Experiment statistical significance**

   Question: Does the paper report error bars suitably and correctly defined or other appropriate information about the statistical significance of the experiments?

   Answer: [Yes]

   Justification: We report the standard deviation for both CMOS and SMOS scores in the main results table (Section 6), reflecting the consistency of human evaluations across participants.

   Guidelines:

   - The answer NA means that the paper does not include experiments.
   - The authors should answer "Yes" if the results are accompanied by error bars, confidence intervals, or statistical significance tests, at least for the experiments that support the main claims of the paper.
   - The factors of variability that the error bars are capturing should be clearly stated (for example, train/test split, initialization, random drawing of some parameter, or overall run with given experimental conditions).
   - The method for calculating the error bars should be explained (closed form formula, call to a library function, bootstrap, etc.)
   - The assumptions made should be given (e.g., Normally distributed errors).
   - It should be clear whether the error bar is the standard deviation or the standard error of the mean.
   - It is OK to report 1-sigma error bars, but one should state it. The authors should preferably report a 2-sigma error bar than state that they have a 96% CI, if the hypothesis of Normality of errors is not verified.
   - For asymmetric distributions, the authors should be careful not to show in tables or figures symmetric error bars that would yield results that are out of range (e.g. negative error rates).
   - If error bars are reported in tables or plots, The authors should explain in the text how they were calculated and reference the corresponding figures or tables in the text.

8. **Experiments compute resources**

   Question: For each experiment, does the paper provide sufficient information on the computer resources (type of compute workers, memory, time of execution) needed to reproduce the experiments?

   Answer: [Yes]

Justification: Justification: We specify the compute resources used for all experiments in Section 6 of the main paper. All models were trained and evaluated using a single NVIDIA RTX 4090 GPU with 24 GB memory. Each experiment was run for 100 epochs with a batch size of 16.

Guidelines:

- The answer NA means that the paper does not include experiments.
- The paper should indicate the type of compute workers CPU or GPU, internal cluster, or cloud provider, including relevant memory and storage.
- The paper should provide the amount of compute required for each of the individual experimental runs as well as estimate the total compute.
- The paper should disclose whether the full research project required more compute than the experiments reported in the paper (e.g., preliminary or failed experiments that didn't make it into the paper).

9. **Code of ethics**

Question: Does the research conducted in the paper conform, in every respect, with the NeurIPS Code of Ethics https://neurips.cc/public/EthicsGuidelines?

Answer: [Yes]

Justification: This research fully conforms to the NeurIPS Code of Ethics and complies with the ethical principles of the Declaration of Helsinki. All data involving human participants were collected under the approval of the institutional Ethics Review Board (ERB) at the affiliated medical center. Written informed consent was obtained from each participant prior to any surgical or recording procedures. The research protocol was reviewed and approved to ensure participant safety, data privacy, and ethical integrity. All sEEG data were anonymized prior to analysis to safeguard confidentiality.

Guidelines:

- The answer NA means that the authors have not reviewed the NeurIPS Code of Ethics.
- If the authors answer No, they should explain the special circumstances that require a deviation from the Code of Ethics.
- The authors should make sure to preserve anonymity (e.g., if there is a special consideration due to laws or regulations in their jurisdiction).

10. **Broader impacts**

Question: Does the paper discuss both potential positive societal impacts and negative societal impacts of the work performed?

Answer: [Yes]

Justification: Our work has potential positive societal impact in the development of neural speech prostheses for individuals with speech impairments, such as patients with aphasia or locked-in syndrome. By enabling speech reconstruction from passive auditory perception, our method may contribute to assistive communication technologies that do not rely on overt articulation or motor function.

Guidelines:

- The answer NA means that there is no societal impact of the work performed.
- If the authors answer NA or No, they should explain why their work has no societal impact or why the paper does not address societal impact.
- Examples of negative societal impacts include potential malicious or unintended uses (e.g., disinformation, generating fake profiles, surveillance), fairness considerations (e.g., deployment of technologies that could make decisions that unfairly impact specific groups), privacy considerations, and security considerations.
- The conference expects that many papers will be foundational research and not tied to particular applications, let alone deployments. However, if there is a direct path to any negative applications, the authors should point it out. For example, it is legitimate to point out that an improvement in the quality of generative models could be used to generate deepfakes for disinformation. On the other hand, it is not needed to point out that a generic algorithm for optimizing neural networks could enable people to train models that generate Deepfakes faster.

- The authors should consider possible harms that could arise when the technology is being used as intended and functioning correctly, harms that could arise when the technology is being used as intended but gives incorrect results, and harms following from (intentional or unintentional) misuse of the technology.
- If there are negative societal impacts, the authors could also discuss possible mitigation strategies (e.g., gated release of models, providing defenses in addition to attacks, mechanisms for monitoring misuse, mechanisms to monitor how a system learns from feedback over time, improving the efficiency and accessibility of ML).

11. **Safeguards**

    Question: Does the paper describe safeguards that have been put in place for responsible release of data or models that have a high risk for misuse (e.g., pretrained language models, image generators, or scraped datasets)?

    Answer: [NA]

    Justification: Our study does not involve any publicly scraped datasets or generative models with a high risk of misuse. The sEEG data were collected from clinical participants who underwent electrode implantation based on their medical needs, as determined by licensed physicians. The entire data collection procedure was reviewed and approved by both the hospital and the institutional ethics review board. All shared data are fully anonymized and used solely for academic research purposes.

    Guidelines:
    - The answer NA means that the paper poses no such risks.
    - Released models that have a high risk for misuse or dual-use should be released with necessary safeguards to allow for controlled use of the model, for example by requiring that users adhere to usage guidelines or restrictions to access the model or implementing safety filters.
    - Datasets that have been scraped from the Internet could pose safety risks. The authors should describe how they avoided releasing unsafe images.
    - We recognize that providing effective safeguards is challenging, and many papers do not require this, but we encourage authors to take this into account and make a best faith effort.

12. **Licenses for existing assets**

    Question: Are the creators or original owners of assets (e.g., code, data, models), used in the paper, properly credited and are the license and terms of use explicitly mentioned and properly respected?

    Answer: [Yes]

    Justification: All data used in our paper were collected by the authors from clinical participants under informed consent and ethical approval. Participants signed a data usage agreement specifying that the data may be used for academic research purposes. The released dataset, NeuroListen, is made publicly available under the **CC BY 4.0** license. We do not rely on any third-party datasets that require separate licenses.

    Guidelines:
    - The answer NA means that the paper does not use existing assets.
    - The authors should cite the original paper that produced the code package or dataset.
    - The authors should state which version of the asset is used and, if possible, include a URL.
    - The name of the license (e.g., CC-BY 4.0) should be included for each asset.
    - For scraped data from a particular source (e.g., website), the copyright and terms of service of that source should be provided.
    - If assets are released, the license, copyright information, and terms of use in the package should be provided. For popular datasets, `paperswithcode.com/datasets` has curated licenses for some datasets. Their licensing guide can help determine the license of a dataset.
    - For existing datasets that are re-packaged, both the original license and the license of the derived asset (if it has changed) should be provided.

- If this information is not available online, the authors are encouraged to reach out to the asset's creators.

13. **New assets**

    Question: Are new assets introduced in the paper well documented and is the documentation provided alongside the assets?

    Answer: [Yes]

    Justification: We introduce a new dataset, NeuroListen, consisting of sEEG-audio paired recordings from five clinical participants. The dataset was collected under clinical supervision with full ethical approval and written informed consent from all participants. All data are anonymized prior to release and accompanied by detailed documentation, including data structure, collection protocol, preprocessing pipeline, license (CC BY 4.0), and usage instructions. At submission time, the dataset is made available through an anonymized repository link.

    Guidelines:

    - The answer NA means that the paper does not release new assets.
    - Researchers should communicate the details of the dataset/code/model as part of their submissions via structured templates. This includes details about training, license, limitations, etc.
    - The paper should discuss whether and how consent was obtained from people whose asset is used.
    - At submission time, remember to anonymize your assets (if applicable). You can either create an anonymized URL or include an anonymized zip file.

14. **Crowdsourcing and research with human subjects**

    Question: For crowdsourcing experiments and research with human subjects, does the paper include the full text of instructions given to participants and screenshots, if applicable, as well as details about compensation (if any)?

    Answer: [Yes]

    Justification: We introduce a new dataset, NeuroListen, consisting of sEEG-audio paired recordings. Prior to data collection, all potential participants were provided with detailed explanations of the experimental procedures, risks, and intended research purposes. Those who agreed to participate signed a formal written informed consent approved by the hospital's ethics committee. The released dataset is fully anonymized and accompanied by comprehensive documentation, including data format, acquisition protocol, preprocessing steps, and licensing (CC BY 4.0). An anonymized access link is included for review.

    All participants received appropriate financial compensation for their time and contribution to the study (200RMB each hour).

    Guidelines:

    - The answer NA means that the paper does not involve crowdsourcing nor research with human subjects.
    - Including this information in the supplemental material is fine, but if the main contribution of the paper involves human subjects, then as much detail as possible should be included in the main paper.
    - According to the NeurIPS Code of Ethics, workers involved in data collection, curation, or other labor should be paid at least the minimum wage in the country of the data collector.

15. **Institutional review board (IRB) approvals or equivalent for research with human subjects**

    Question: Does the paper describe potential risks incurred by study participants, whether such risks were disclosed to the subjects, and whether Institutional Review Board (IRB) approvals (or an equivalent approval/review based on the requirements of your country or institution) were obtained?

    Answer: [Yes]

Justification: This study involved human participants undergoing clinical stereo-electroencephalography (sEEG) procedures. All participants were recruited from a certified hospital and were provided with detailed explanations of the experimental procedures, potential risks, and data usage plans. Written informed consent was obtained from each participant prior to data collection. The entire study protocol was reviewed and approved by the Institutional Review Board (IRB) and the affiliated hospital ethics committee, in full compliance with national and institutional ethical regulations.

Guidelines:

- The answer NA means that the paper does not involve crowdsourcing nor research with human subjects.
- Depending on the country in which research is conducted, IRB approval (or equivalent) may be required for any human subjects research. If you obtained IRB approval, you should clearly state this in the paper.
- We recognize that the procedures for this may vary significantly between institutions and locations, and we expect authors to adhere to the NeurIPS Code of Ethics and the guidelines for their institution.
- For initial submissions, do not include any information that would break anonymity (if applicable), such as the institution conducting the review.

16. **Declaration of LLM usage**

Question: Does the paper describe the usage of LLMs if it is an important, original, or non-standard component of the core methods in this research? Note that if the LLM is used only for writing, editing, or formatting purposes and does not impact the core methodology, scientific rigorousness, or originality of the research, declaration is not required.

Answer: [No]

Justification: Large language models (LLMs) were not used in the core methodology.

Guidelines:

- The answer NA means that the core method development in this research does not involve LLMs as any important, original, or non-standard components.
- Please refer to our LLM policy (`https://neurips.cc/Conferences/2025/LLM`) for what should or should not be described.

