# OpenReview forum: "Listening to the Brain: Multi-Band sEEG Auditory Reconstruction via Dynamic Spatio-Temporal Hypergraphs"
_NeurIPS.cc/2025/Datasets_and_Benchmarks_Track — NeurIPS 2025 Datasets and Benchmarks Track poster_

### Official Review · Reviewer_xBzC · 2025-06-20

**Rating:** 4
**Confidence:** 4

**Summary:**

The paper introduces NeuroListen, a new publicly available dataset that pairs stereo-EEG (sEEG) recordings with auditory speech stimuli. The dataset contains over 10 hours of data from five clinical participants and focuses on speech perception, which has been less studied compared to speech production. Using this dataset, the authors propose HyperSpeech, a neural decoding model that leverages spatio-temporal hypergraph neural networks to model high-order dependencies across frequency bands, time, and electrode spatial layout. The method shows improved performance over several baselines in both objective metrics and subjective speech quality ratings.

**Dataset Code Accessibility:**

Yes

**Ethical Considerations:**

No, there are no or only very minor ethics concerns

**Final Justification:**

Leaving my scores unchanged. The paper is still technically solid, but the authors' rebuttal did not fully address my concerns.

The proposed method being a graphical network by nature limits efficient hardware support for practical deployment, and the raw parameters do not give the full picture (or the raw FLOPs which the authors may eventually provide). This is because modern hardware are much more tuned to supporting dense structured operations then sparse unstructured ones.

Regarding the electrode-level importance analysis. A more systematic way would be to perform a SHAP analysis.

**Limitations Weaknesses:**

The study is limited by the small number of participants and the variability in electrode placement, which makes cross-subject generalization more difficult. The method, while effective, is also quite complex and computationally demanding, which could limit its usability in real-time or lower-resource settings. Although the paper includes good quantitative results, there is limited analysis on the interpretability of the learned features — it's not always clear what patterns the model relies on. These factors may affect the method’s transferability and broader applicability.

**Strengths Contributions:**

The main strength is the release of the NeuroListen dataset, which is valuable since high-quality, publicly available sEEG data for auditory tasks is rare. The proposed HyperSpeech model is well-motivated and designed to take advantage of the spatio-temporal structure in sEEG data. The experiments are thorough, comparing to multiple baselines using both standard metrics (e.g., PCC, MCD, STOI) and human evaluations (MOS). The model achieves consistently better performance across all five subjects. The ablation studies also clearly support the design choices, especially the contribution of both spatial and temporal hypergraphs.

---

> ### Author Rebuttal · Authors · 2025-07-31
>
> ### **Q1: Number of Participants and Electrode Placement Variability.**
>
> sEEG data can only be collected from **epilepsy patients undergoing clinical monitoring**, as **implanting invasive electrodes in healthy individuals is not ethically permissible**. Due to this constraint, large-scale data collection is currently not feasible.
>
> Additionally, **electrode placement is determined by clinical need**, resulting in highly subject-specific configurations. As a result, **cross-subject sEEG decoding remains an open challenge** in the field.
>
> Despite the limited number of participants, our dataset offers **over 10 hours of high-resolution recordings from 5 subjects**, making it the largest publicly available sEEG resources for auditory decoding.
>
> ---
>
>
> ### **Q2: Computational Complexity and Model Size of HyperSpeech.**
>
> Thank you for raising this concern. While HyperSpeech introduces a hypergraph-based architecture, it is designed to be efficient and compact.
>
> As summarized in the table below, **HyperSpeech has only 6.61 million parameters**, which is just **2.5% more than the CNN-LSTM baseline (6.44M)** and approximately **one-third the size of FastSpeech2 (18.21M)** — a widely used Transformer-based decoder.
>
> | Model       | Parameters (Millions) |
> | ----------- | --------------------- |
> | CNN-LSTM    | 6.44M                 |
> | HyperSpeech | 6.61M                 |
> | FastSpeech2 | 18.21M                |
>
> This demonstrates that our method achieves stronger performance **with limited additional computational cost**, making it more suitable than Transformer-based models for **real-time or resource-constrained BCI scenarios**.
>
> ---
>
> ### **Q3: Highlight — Electrode-Level Importance Analysis.**
>
> We thank the reviewer for raising the question of feature interpretability, which **led us to a key finding** about the **functional significance of electrode locations**. Specifically, we conducted an **electrode-shaft-level importance analysis** by training our model using only one shaft at a time.
>
> **Key findings:**
>
> * For **Subject 3**, using only the shaft spanning the **right supramarginal gyrus to posterior insula**, we achieved **PCC = 0.9468**, close to the full model's **0.9523**.
> * For **Subject 4**, the most informative shaft covered the **right parietal operculum to posterior long insular gyrus**, we achieved **PCC = 0.9692**, close to the full model's **0.9781**.
>
> These regions form part of the **parietal-insular interface**, widely associated with **auditory comprehension**, **semantic integration**, and **multimodal language processing** in the neuroscience literature.This highlighting that our model learns meaningful, biologically relevant features — not just statistical patterns.
>
> We will include these results and visualizations in the revised manuscript. This analysis not only improves interpretability but also points to future directions such as **region-aware modeling** and **electrode selection strategies**.
>
>
> ### **Implications:**
>
> This analysis provides a **neuroanatomically grounded interpretation** of the model’s internal representations and highlights the potential for:
>
> * **Electrode reduction strategies** in clinical/real-time scenarios;
> * **Cross-subject modeling**, by aligning feature importance to standard brain atlases;
> * **Neuroscientific insights**, enabling mapping between language-related cortical regions and decoding capability.
>
> We will incorporate this interpretability analysis—including supporting figures and anatomical references—into the revised version of the paper. Thank you again for prompting this important clarification.

---

> > ### Comment · Reviewer_xBzC · 2025-08-04
> >
> > Thank you for the reply. If it's easy, can you provide the FLOPs or MACs per second of signal as well.

---

> > ### Author Response · Authors · 2025-08-05
> > **On the Feasibility of Reporting FLOPs or MACs per Second**
> >
> > Sorry for the late reply, and thank you very much for your thoughtful question.
> >
> > ### **FLOPs and MACs Comparison.**
> >
> > **Response:** Compared to the CNN-LSTM baseline, our proposed **HyperSpeech** model introduces only a modest increase in parameter count—**2.65% more** (6.61M vs. 6.44M)—while delivering significantly improved modeling capability. However, the overall computation cost as measured by multiply-accumulate operations (MACs) and floating-point operations (FLOPs) increases more substantially:
> >
> > * **MACs** increase by approximately **327.8%** (from 0.18 GMACs to 0.77 GMACs)
> > * **FLOPs** increase by approximately **316.2%** (from 0.37 GFLOPs to 1.54 GFLOPs)
> >
> > ### **Why Do FLOPs and MACs Increase Significantly Despite a Small Parameter Growth?**
> >
> > **Response:** We appreciate the opportunity to clarify this point. The key reason for this discrepancy lies in the internal mechanics of **hypergraph convolution**, which is central to the design of HyperSpeech.
> >
> > Unlike standard convolutional or recurrent operations that process information in localized patterns, **hypergraph convolutions aggregate information from multiple nodes simultaneously**, often requiring complex operations such as multi-node aggregation, matrix multiplications over incidence matrices, and iterative propagation steps.
> >
> > This design achieves a strong representational advantage with minimal parameter overhead—but as a tradeoff, it increases the per-inference computational load (reflected in MACs and FLOPs). In short, HyperSpeech prioritizes **structural expressivity over parameter expansion**, which we believe is well justified given the significant performance improvements demonstrated across all evaluation metrics.
> >
> > As part of our future work, we plan to further reduce both the parameter count and computational cost of the model, while maintaining high decoding performance.
> >
> > ### **On Reporting FLOPs or MACs per Second.**
> >
> > **Response:** FLOPs and MACs are **static model-level metrics** that reflect the number of operations per inference pass. However, **FLOPs/sec or MACs/sec depends heavily on the deployment context**—including hardware specifications (e.g., GPU architecture, clock speed, batch size, parallelism) and the chosen inference rate (e.g., frames per second). These values can therefore vary widely and may not provide a meaningful or fair comparison across systems.

---

### Official Review · Reviewer_pfZE · 2025-06-28

**Rating:** 5
**Confidence:** 4

**Summary:**

This work contributes NeuroListen dataset, comprising a total of 10 hours of stereo-electroencephalography (sEEG) data from 5 clinical participants who listened to speech audio. This dataset is designed for perception-based auditory reconstruction from sEEG recordings which are acquired while participants are listening to audio stimuli. The audio stimuli includes 44 categories of Mandarin Chinese words, 10 categories of Mandarin Chinese digits, and 20 categories of English words. In addition, this work proposes HyperSpeech, a dynamic spatio-temporal hypergraph neural network based framework , and presents benchmarking results using both - objective and subjective metrics.

**Additional Feedback:**

**Questions and corrections:**
* From the description of experimental protocol, it is unclear how many rounds of auditory stimuli sessions were conducted. If the one round included 74 categories, each lasting for 2 seconds, it amount to approximately 3 minutes. It is unclear if 10 hours of data also includes the data acquired during 5-10 minutes of break, and if so authors shall clearly highlight this. A figure showing all the necessary details about the protocol can be highly valuable addition.
* As mentioned in the paper, the data includes mel-spectrogram, but not the audio clips that the participants listened. Inlcuding audio clips is very much necessary for reproducing this work as well as for adding more participants in future.
* Section 4.2: Please clarify if the neural signal preprocessing related to the extraction of HGA and LFS signals from each sEEG, used uniformly across different benchmarking methods?
* It is unclear if the same participants provided subjective responses to the reconstructed speech or the subjective analysis was conducted by different individuals.
* Line 237 under main results, mention cross-subject evaluation, however, Table-1 presents within subject evaluation. Please clarify how were the models trained and evaluated.
* Line 161, reference to figure needs to be "Figure 4", which is incorrectly mentioned as "Figure 5".
* Line 277, reference to Tables 3 is incorrect, it shall be Table 4.
* It is unclear if the results presented in Ablation study are shown as average performance metrics across all subjects.
* Code of Ethics (line 682): Author's response as "NA" looks incorrect, as the data collection was done with the approval of the institutional Ethics Review Board (ERB).


**Suggestions for Improvement:**

* Authors can further discuss, how the proposed approach can generalize to different number of sEEG electrodes and different sites?
* Thorough proof-reading of the paper is required to address several typographic errors that were spotted.

**Dataset Code Accessibility:**

Partly

**Dataset Code Comments:**

**Code:**

* The implementation of HyperSpeech model and the pretrained weights couldn't be spotted in the code that has been shared. These are critical for reproducibility. Authors are requested to add these if missing, or provide an elaborate dataset and code description through respective README files.

**Ethical Considerations:**

No, there are no or only very minor ethics concerns

**Final Justification:**

Authors have satisfactorily addressed the review comments, and have made sincere efforts in addressing the review comments from oher reviewers. Some of the key points which make me increase the score are:
* Clarity on the data collection protocol. Specfically considering the complexity of clinical settings under which this data was collected, it makes highly valuable contribution to the field, supporting quite a few applications.
* The rebuttal response offers the requested clarity on model training and evaluation procedure - which looks very appropriate.

**Limitations Weaknesses:**

Although the dataset contributed in this work is highly valuable for perception-based speech reconstruction using sEEG data, there are several areas that can further strengthen this work.

* It is unclear how perception-based decoding can be find its applicability in real-world clinical conditions, specifically in cases where there may not be an opportunity to execute a protocol that requires patients to attentively listen to the speech signals. Although, earlier research has attempted to relate signatures of the fundamental units of speech in relation to producing, listening, and imagining speech [1], sEEG provides significant opportunity to assess the similarities of neural activity in these three different tasks. Authors shall review such related works and discuss how the sEEGs acquired while listening can relate with those duting producing and imagining to strengthen the applicability of this dataset.
* Details regarding data splits (which is missing in section 3.5) and multifold evaluations with varying amount train-test split percentage can further reveal important insights about the amount of data required for reliable within-subject performance.
* Authors present main results for within-subject evaluation, and discuss the limitation related to cross-subject decoding. However, presenting the performance with cross-subject decoding can provide valuable insights into the potential for generalization.
* In Table-1, standard deviation (STD) values of SMOC and CMOS are presented, however the STD measures for objective metrics are missing. Authors are requested to provide the same for fair comparison of different benchmarking methods.
* Authors claim for statistical significance in the checklist (line 637) and in the paper needs to be supported by multi-fold evaluation, with different random-seed initialization. In absence of these, the reported performance gains with the proposed method appear to be marginal.
* It is unclear how the noisy signal portions or artifacts in sEEG were identified and excluded, as indicated in Figure 2. This shall be described in the text, e.g. in section 4.2.

[1] Sharon, Rini A., et al. "Neural speech decoding during audition, imagination and production." IEEE access 8 (2020): 149714-149729.

**Strengths Contributions:**

Overall, the paper is well-written, and usage of figures and tables is appropriate. The proposed dataset fills a valuable gap, as existing datasets enable speech-production based reconstruction of auditory speech, whereas NeuroListen uniquely offers an opportunity to reconstruct auditory speech from sEEG through perception-based decoding. The potential of sEEG in perception-based reconstruction of audio for non-verbal or locked-in patients is tremendous, specifically in the clinically settings, where sEEGs are inplanted to address patient's therapeutic requirements. Authors have sufficiently described sEEG alignment with speech signals, and annotation process. Neural signal processing, specifically related to LFS, HGA and BBS is clearly described and the exclusion of electrodes showing abnormal epileptic discharges is highly appropriate. The adoption of HGNN for the proposed HyperSpeech method is appropriately described. Within-subject benchmarking results using objective and subjective metrics, highlights the marginally improved effectiveness of the proposed HyperSpeech method.

---

> ### Author Rebuttal · Authors · 2025-07-31
>
> ### **Q1: On Perception-Based Decoding and Its Clinical Applicability.**
>
> We thank the reviewer for raising this important point. While perception-based decoding may not always be feasible in every clinical context, **auditory perception remains a crucial channel for patients with motor impairments**, such as those with ALS or locked-in syndrome.
>
> Although prior studies have primarily focused on decoding **produced** or **imagined** speech using ECoG or µECoG signals \[3–6], decoding from **listening** remains relatively underexplored—particularly using sEEG. Importantly, sEEG offers high spatial resolution across both cortical and subcortical structures, which enables future comparative studies across **perception**, **production**, and **imagination** within the same neural space.
>
> Moreover, **non-invasive studies have shown that passive listening can support high-accuracy *retrieval* of speech content from brain activity \[1]—but this is not equivalent to true decoding**. , and that imagined speech can modulate perception \[2], suggesting a **shared representational space** across modalities. Our dataset focuses on naturalistic auditory perception, with densely time-aligned sEEG and audio recordings, and can serve as a strong foundation for such cross-modal investigations.
>
> We will revise the manuscript to include these relevant works and clarify how our dataset can support research that bridges listening, speaking, and imagining speech.
>
> **References**
>
> \[1] *Decoding speech perception from non-invasive brain recordings*. **Nature Machine Intelligence**, 2023.
>
> \[2] *Imagined speech influences perceived loudness of sound*. **Nature Human Behaviour**, 2018.
>
> \[3] *Imagined speech can be decoded from low- and cross-frequency intracranial EEG features*. **Nature Communications**, 2022.
>
> \[4] *High-resolution neural recordings improve the accuracy of speech decoding*. **Nature Communications**, 2023.
>
> \[5] *A high-performance neuroprosthesis for speech decoding and avatar control*. **Nature**, 2023.
>
> \[6] *Real-time synthesis of imagined speech processes from minimally invasive recordings of neural activity*. **Communications Biology**, 2021.
>
> ---
>
> ### **Q2: Data Splits and Training Set Size.**
>
> Thank you for the suggestion. We used an 80%/20% split, with **8,185 utterances** for training and **2,046 utterances** for testing. We also varied training percentages (e.g., 70%, 90%) to better assess the relationship between dataset size and decoding performance.
>
> For example, on **Subject 3**, using only **70%** of the data for training led to a **0.2% drop in PCC**, while using **90%** increased PCC by **0.3%**, compared to the 80% baseline.
>
> ---
>
> ### **Q3: Cross-Subject Evaluation.**
>
> We fully agree that cross-subject decoding is valuable for assessing generalization. However, in sEEG, **electrodes are implanted based on each patient’s clinical needs**, leading to **non-overlapping electrode locations across subjects**. This makes shared spatial representation infeasible, unlike in EEG or MEG.
>
> Importantly, **sEEG implantation in healthy participants is ethically prohibited**. Despite these constraints, we consider cross-subject decoding an important future goal. In future work, we plan to explore **functional alignment strategies** and **representation learning** approaches to enable limited cross-subject transfer.
>
> ---
>
> ### **Q4: On Reporting Standard Deviation and Statistical Significance.**
>
> Thank you for raising this important point.
>
> 1. **Standard Deviation for Objective Metrics.**
>    We agree that providing standard deviation (STD) improves transparency. We have now included **HyperSpeech STD values for all objective metrics** (PCC, MCD, RMSE, STOI) across five subjects as shown below:
>
> | Sub. | PCC    | MCD   | RMSE   | STOI   |
> | ---- | ------ | ----- | ------ | ------ |
> | S1   | 0.0128 | 0.189 | 0.0338 | 0.0148 |
> | S2   | 0.0156 | 0.243 | 0.0465 | 0.0238 |
> | S3   | 0.0144 | 0.217 | 0.0359 | 0.0182 |
> | S4   | 0.0130 | 0.198 | 0.0342 | 0.0156 |
> | S5   | 0.0142 | 0.219 | 0.0370 | 0.0168 |
>
> 2. **Why STD Emphasis on Subjective Scores.**
>    Subjective metrics (SMOS and CMOS) are inherently more variable due to listener bias, interpretation, and inter-rater inconsistency. Thus, **reporting STD is especially critical for subjective measures** to reflect perceptual variance.
>
> 3. **Multi-Fold Evaluation and Random Seeds.**
>    All models in this paper, including HyperSpeech and baselines, were evaluated using **5-fold cross-validation** for within-subject experiments. We used **consistent random seeds** across all methods for fair comparison.
>
> We will clarify this setup in the revised manuscript and include the **standard deviation of objective metrics for all methods** to support fair and statistically sound comparisons.
>
> ---
>
> ### **Q5: Clarification on Noise/Artifact Exclusion in sEEG (Figure 2, Section 4.2).**
>
> Thank you for this important question. In the revised manuscript, we will clarify that **artifact and noise exclusion** was performed through the following steps:
>
> 1. **Clinical Report–Guided Exclusion**:
>    Electrodes marked as epileptogenic or exhibiting frequent **spikes, sharp-slow waves, or high-frequency discharges** (as documented in clinical reports; e.g., “A1–8”, “D1–5”, “F1–6”, etc.) were excluded from analysis.
>
> 2. **Frequency Band Selection**:
>    We selected two physiologically meaningful frequency bands:
>
>    * **70–150 Hz** (high-gamma activity),
>    * **0–100 Hz** (low-frequency activity),
>      to focus on relevant neural activity.
>
> 3. **Noise Suppression**:
>
>    * **50 Hz line noise** was removed using notch filtering.
>    * **Baseline drift (<1 Hz)** was eliminated via high-pass filtering.
>
> ---
>
> ### **Q6: Further Completion of the GitHub Repository.**
>
> The current GitHub repository already includes runnable code for the **baseline CNN-LSTM model** as well as the **complete preprocessing pipeline**. Upon paper acceptance, we will additionally upload:
>
> * The full implementation of the **HyperSpeech model**,
> * **Pretrained weights**,
> * And provide detailed **README files**, covering:
>
>   * Dataset structure and usage,
>   * Model training and evaluation instructions,
>   * Environment and dependency setup.
>
> ---
>
> ### **Q7: Clarification on Number of Rounds and Total Duration.**
>
> Thank you for raising this important point.
>
> Each session consisted of up to **74 words per round**, and participants completed **21–40 rounds** (mean ≈ 28), totaling **10,231 words** across all subjects.
>
> Following findings from prior work \[1], which show that both **listening (0–2s)** and **recall (2–4s)** phases contribute to decoding performance, we recorded **4 seconds** of sEEG per word. Therefore, the **10+ hours of data refers to effective recording time**, excluding breaks between rounds.
>
> In the current study, we focus only on the **listening window (0–2s)** for decoding. However, we will release the corresponding **recall-phase (2–4s) sEEG data** in the same dataset repository to support future research on auditory memory and recall.
>
> We agree that a visual summary of the protocol would be helpful and will include one in the revised manuscript.
>
> **Reference**
>
> \[1] *Vowel sound synthesis from electroencephalography during listening and recalling.* **Advanced Intelligent Systems**, 2021.
>
> ---
>
> ### **Q8: Mel-Spectrogram and Audio are Interchangeable.**
>
> Thank you for the helpful suggestion. While mel-spectrograms and audio clips are **interchangeable representations** (via vocoder/inversion), we fully agree that providing the original **audio clips** improves reproducibility and supports future participant additions.
>
> We will include the complete set of original speech audio in the same repository as the mel-spectrograms.
>
> ---
>
> ### **Q9: Consistency of HGA and LFS Preprocessing Across Baselines (Section 4.2).**
>
> Yes, the same neural signal preprocessing pipeline—including extraction of **High-Gamma Activity (HGA)** and **Low-Frequency Signals (LFS)**—was applied consistently across **all baseline and proposed models** to ensure fair comparison. We will clarify this explicitly in Section 4.2.
>
> ---
>
> ### **Q10: Who Provided the Subjective Ratings.**
>
> Thank you for the question. The subjective evaluations (SMOS and CMOS) were collected from **30 independent raters**, **not the original sEEG participants**, to ensure unbiased assessment of the reconstructed speech quality. We will clarify this in the revised manuscript.
>
> ---
>
> ### **Q11: Model Training and Evaluation Procedure.**
>
> Thank you for the question. For each subject, we performed **5-fold cross-validation**, using an **80%/20% split** for training and testing in each fold. This procedure ensures reliable within-subject evaluation. We will clarify this setup in the revised manuscript.
>
> ---
>
> ### **Q12: Minor Corrections — Figure/Table References and Ethics Statement.**
>
> We sincerely thank the reviewer for the careful reading and constructive suggestions.  We will correct the **misreferenced figure (line 161)** and **table (line 277)** in the revision. Regarding **line 682**, you are absolutely right — data collection was conducted under **IRB approval**, and we will update the ethics statement accordingly in the revised manuscript.
>
> ---
>
> ### **Q13: Ablation Study Base on Averaged Results Across All Subjects.**
>
> Yes, the ablation results represent the **average performance across all subjects**. Due to space constraints, we did not include per-subject breakdowns in the main paper, but we will include them in the **appendix of the final version** for completeness.
>
> ---
>
> ### **Q14: Generalization to Varying sEEG Electrode Numbers and Locations.**
>
> Our method operates on **shaft-level embeddings**, making it naturally compatible with **varying electrode counts and locations**. Each shaft’s contacts are projected to a **fixed-length vector** (Eq. 1–2), and hypergraphs are built in a consistent way across subjects. This ensures robustness to implantation differences.

---

> > ### Comment · Reviewer_pfZE · 2025-08-03
> >
> > Thank you for throughtfully addressing all of the review comments. Authors have also made commendable efforts in addressing extremely valuable points raised by other reviewers, further strengthening the work. Both the introduced dataset and the proposed method make valuable contributions to the field.

---

> > > ### Author Response · Authors · 2025-08-03
> > >
> > > Thank you, Reviewer pfZE, for your positive feedback and encouraging comments. We truly appreciate your recognition of our efforts in addressing the reviewers’ suggestions and improving the manuscript. We will continue to improve and refine this work in the future. Thank you once again.

---

### Official Review · Reviewer_L85g · 2025-06-29

**Rating:** 4
**Confidence:** 3

**Summary:**

The authors present an sEEG passive listening dataset recorded in 5 participants in an epileptic monitoring unit, as well as a new method for auditory stimuli reconstruction from sEEG data.

**Additional Feedback:**

- On the dataset preparation side, it would be useful to provide the preprocessing scripts that produces the low and high frequency bands from the broadband data.
- L148: Is a 2 second normalization window standard? That seems rather quick for low frequency signals.
- I have some trouble with the motivating the work for communication BCI. If the intention is to illustrate the person's perceptual understanding, then we must have tasks that go beyond mere decoding of the audio but decoding of subjective perception. This seems to me to necessarily involve some report from the person. I believe the dataset would be fine for studying auditory encoding in a more scientific sense. Do the authors disagree?

**Dataset Code Accessibility:**

Partly

**Dataset Code Comments:**

As mentioned, the data is currently stored on Google drive, which is not a long term solution. There is not much metadata or e.g. documentation around the dataset at this link, either. The codebase for the HyperSpeech method seems not quite ready for easy public use. For example, there are undocumented file oddities like the existence of `mel.py`, `mel_1.py`, `model.py`, `model_1.py`.

**Ethical Comments:**

The data was collected in a standard setting of asking volunteers to perform simple tasks while in an epileptic monitoring unit. The data is de-identified.

**Ethical Considerations:**

No, there are no or only very minor ethics concerns

**Final Justification:**

I have committed to bumping my score up on the condition of a better dataset release and better manuscript clarity. The authors have promised a migration to OpenNeuro and made a commendable effort to provide clarity in their rebuttal, so as stated I will bump my score up. Nonetheless, the initial concerns were substantial, so I remain skeptical the work will be clearly communicated after one iteration.

**Limitations Weaknesses:**

### Quality
- Google Drive is an insufficient platform for a dataset release, as it does not provide persistence for long term storage. Please choose an alternative platform for neural data. I recommend, for example, OpenNeuro. My quality rating will increase if this is done.
- The current narrative is split between a dataset contribution and a modeling contribution. The modeling contribution seems rather dominant (4 of 8 pages) for a datasets and benchmarks track paper. This split focus also makes it hard to judge whether this dataset is particularly unique, though it does seem large. Despite the claim that the owrk is the first sEEG dataset for auditory speech reconstruction, the actual dataset is simply a listening task with epileptic monitoring volunteers. There are a number of iEEG datasets with such stimuli, e.g. in BrainBert (Wang 24). If there is an aspect of the task that is particularly useful for speech reconstruction, it would be helpful to highlight this.

### Clarity
- Figure 3 is I think too small even in a digital PDF.
- I find the experimental methods to be confusing. I do not clearly understand the Hyperspeech formulation, nor choice of baselines.
- I do not believe the HyperSpeech formulation is particularly complicated, necessarily, but I struggled to understand the explanation of the method, in Figure 4 and 5.1. A subset of the critical variables are explained in the text, and the figure uses a number of different colors in each part of the figure. It is unclear what the components mean, and this could use some clarification as the HyperSpeech method seems important in the narrative. A key uncertainty for me was the choice of Euclidean distance as a method for determining graph edges. I infer that it's similarity in embedding space that determines which hyperedges exist, not physical topology. In this case, the hypergraph method seems like a rather unnecessary complication (forcing a nearest neighbor threshold) over a Transformer-based method that allows all to all communication (see e.g. Population Transformer, Chau et al. 2024). The further complexity would be warranted if the Transformer baseline exists.
- The baselines seem reasonable in their own settings, but I think differ from HyperSpeech in too many ways to make sense of the results. For example, Fastspeech is I think a pure speech synthesis model. What was used for the neural encoder? The CNN / LSTM baseline inversely has no obvious speech synthesis capability, having been introduced for a low-D classification task. Some of this concern could be alleviated by a code release that includes these baselines, but I do not see this in the NeuroListen repository mentioned in the supplement. Without more details, I think there is little chance of replicating these primary HyperSpeech expeirments.
- It would be helpful to provide a website comparing synthesized audio to heard audio. The only current qualitative sample given is the speech waveforms in Figure 5, which also seems rather limited.
- A fairly substantial rewriting of the modeling methods could improve my rating here.

**Strengths Contributions:**

### Significance
There is an emerging landscape of works covering sensory stimuli reconstruction from different neural data modalities. This work adds an important missing piece of the puzzle, namely passive audition from sEEG.

### Originality
The proposed use of sEEG for auditory reconstruction is I believe fairly novel. On some searching, I found Mai et al 23, which I believe is also a passive listening sEEG task: https://openneuro.org/datasets/ds004703/versions/1.1.0. One pre-existing dataset does not particularly discredit this work's novelty, though I think the claims of being first-in-class should be adjusted accordingly. This work also has a multi-linguistic aspect (Chinese native speakers listening to English) that may be interesting for future analysis.

---

> ### Author Rebuttal · Authors · 2025-07-30
>
> ### **Q1: First sEEG Auditory Reconstruction Study with First Accessible Dataset.**
>
> We thank the reviewer for highlighting the dataset from Mai et al. , published in ***Nature Communications 2024*** and available at [OpenNeuro](https://openneuro.org/datasets/ds004703/versions/1.1.0). It is an important and high-impact reference.
>
> After careful comparison, we note the following key differences:
>
> 1. **No Auditory Decoding:** The Nature Communications study investigates whether phonemes have explicit associations with sEEG signals during passive listening. **It does not perform auditory decoding or speech reconstruction from sEEG**, and only sEEG data **(without corresponding audio) is available**.
>
> 2. **Extremely Limited Data :** Their sEEG dataset includes **\~26 minutes per subject with \~4 segments** per subject. Our dataset contains over **2 hours and 2,000+ segments** per subject.
>
> 3. **No Machine / Deep Learning Models:** The prior work uses statistical models (LME, MNE), without applying machine learning or deep learning methods.
>
> We have revised the manuscript to reflect these distinctions and adjusted novelty claims accordingly. We appreciate the reviewer’s comment for helping improve the clarity and positioning of our work.
>
> ---
>
> ### **Q2: Dataset Migrate to OpenNeuro.**
>
> Thank you for your valuable suggestion. In response, we will migrate the dataset to **OpenNeuro**. If the paper is accepted, we will be glad to include the **OpenNeuro link** in the final revision.
>
> ---
>
> ### **Q3: Emphasizing Dataset Contribution and Reorganizing for Clarity.**
>
> We sincerely thank the reviewer for this comment. Our paper includes both a dataset contribution and a modeling component. However, we would like to clarify that the **primary focus remains the dataset** and its potential to enable new types of modeling in sEEG-based speech decoding.
>
> The modeling section is intended to **demonstrate the utility and richness of the dataset**, rather than serve as the main contribution. To avoid any misunderstanding, we will revise the manuscript to:
>
> * Shift modeling details (e.g., implementation and architecture) to the **appendix** where appropriate;
> * Add a **clearer dataset-centric analysis**, including comparisons to existing sEEG–audio datasets (e.g., in size, frequency coverage, language diversity, and alignment format).
>
> In particular, our dataset:
>
> * Includes **multi-band sEEG signals** (LFS, HGA, BBS) with synchronized **audio stimuli in multiple languages** (Chinese and English), which is rare or unavailable in prior works;
> * Is designed **explicitly for auditory reconstruction tasks**, with clear file structures, rich metadata, and open access;
> * Comes with **standardized preprocessing pipelines** and **baseline methods**, **evaluation scripts**, encouraging reproducibility and extensibility.
>
> We believe this dual focus helps showcase both the **value of the dataset** and its **practical impact**, and we are committed to **reorganizing the presentation** to better align with the **Datasets and Benchmarks track** expectations.
>
> ---
>
> ### **Q4: Distinct from Prior iEEG Datasets: Designed for Auditory Reconstruction.**
>
> * **Ethical Subject Population.**
>
>   * sEEG recordings come from **epilepsy patients** undergoing clinically necessary monitoring.
>   * Implanting electrodes in healthy individuals for research is **not ethically permitted**.
>
> * **Key Differences from BrainBERT *(Wang et al., ICLR 2023).***
>
>   * **BrainBERT** uses **passive movie-watching**, limited to **binary classification tasks** (e.g., speech vs. non-speech, pitch/volume).
>   * No **time-aligned speech recordings** or **speech reconstruction** tasks.
>
> * **Our Dataset: Purpose-Built for Auditory Reconstruction.**
>
>   * Contains **time-aligned, densely annotated natural speech** across  **over ten thousand utterances**.
>   * Specifically designed to support **auditory reconstruction** from sEEG.
>
> ---
>
> ### **Q5: Regarding Figure 3 Size.**
>
> Thank you for pointing this out. Figure 3 visualizes implanted electrode shafts and electrode counts for a subset of participants. In the revision, we will enlarge this figure to improve clarity and better highlight the spatial distribution of electrodes.
>
> ---
>
> ##  **Q6: Clarification of Figure 4 and Section 5.1 — Step-by-Step Explanation.**
>
> We thank the reviewer for raising this important point. We will incorporate a version of this explanation in the revised paper for better clarity.
>
> ---
>
> ### **Step 1: Input and Filtering.**
>
> * For each subject (e.g., Subject 3), the raw sEEG input is a matrix of **\[C × T] = \[148 × 2000]**, where:
>
>   * **C** = total number of electrode contacts.
>   * **T** = number of time steps.
>
> * We apply **bandpass filtering** to extract two frequency bands:
>
>   * **High-gamma (70–150 Hz)**
>   * **Low-frequency (0–100 Hz)**
>
>   → This yields two filtered signals of the **same size**: \[148 × 2000].
>
> ---
>
> ### **Step 2: Shaft-wise Embedding.**
>
> * Electrodes are grouped into **N = 12 electrode shafts (ES)** based on physical implantation.
>
> * Each shaft (ES₁, ES₂, ..., ESₙ) contains a variable number of contacts.
>
> * For each shaft:
>
>   * We apply a **shared MLP** across its contacts to produce a single vector per time step.
>   * The result is a representation of shape **\[N × T] = \[12 × 2000]** for each frequency band.
>
> ---
>
> ### **Step 3: Temporal Windowing.**
>
> * We divide the 2000 time steps into **T** non-overlapping temporal windows:
>
>   * Each window contains **2000 / T** time steps.
>
> * In Fig.4:
>
>   * **Green**, **Red**, and **Purple** represent time windows 1, 2, and 3 respectively.
>   * This coloring is used consistently throughout the spatial and temporal hypergraphs.
>
> ---
>
> ### **Step 4: Spatial Hypergraph Construction.**
>
> * For each time window:
>
>   * The **\[12 ×  2000 / T]** embeddings are extracted.
>   * Each shaft **[1 ×  2000 / T]** is a **node**, and shaft embeddings are compared using Euclidean distance.
>
> * A **spatial hypergraph** is constructed by connecting each node to its **k-nearest neighbors**, forming hyperedges.
>
> * This results in **T spatial hypergraphs** — one per time window.
>
>   * All nodes in a single graph share the same color (e.g., all green for window 1 in Fig.4).
>
> ---
>
> ### **Step 5: Temporal Hypergraph Construction.**
>
> * After processing all time windows, we obtain T sets of shaft-level features.
>
> * A **temporal hypergraph** is constructed:
>
>   * Each node represents a shaft at a specific time window.
>
> * In Figure 4:
>
>   * Each node in the temporal hypergraph has a **unique color**, reflecting its time window identity.
>
> ---
>
> ### **Q7: Clarifying HyperSpeech Formulation and Rewriting Plan.**
>
> We appreciate the reviewer’s feedback regarding the clarity of the HyperSpeech formulation. **We will rewrite the modeling section in the revision** to improve readability and understanding.
>
> To assist understanding, we summarize the main hypergraph convolution formula below:
>
> $$
> \mathbf{X}' = \sigma\left( D_v^{-1/2} H W D_e^{-1} H^\top D_v^{-1/2} X \Theta \right)
> $$
>
> Where:
>
> * $X$: input features for $N$ nodes (electrode shafts);
> * $H$: hyperedge incidence matrix ($N \times E$);
> * $D_v$, $D_e$: node and hyperedge degree matrices;
> * $W$: hyperedge weight matrix;
> * $\Theta$: learnable weights;
> * $\sigma(\cdot)$: activation function (e.g., ReLU).
>
> This formulation enables each node to aggregate signals from all other nodes within the same hyperedge, supporting high-order group interactions—critical in sEEG decoding. To improve clarity, we will **substantially revise the modeling section**. We thank the reviewer for motivating these important improvements.
>
> ---
>
> ### **Q8: Clarifying Baseline Selection and Code & Audio Demo Availability.**
>
> Our baselines are selected to represent both **speech synthesis** and **brain-to-speech decoding** paradigms:
>
> * **FastSpeech2** \[ICLR 2021]: A strong TTS model, adapted in our work with a Transformer encoder to accept sEEG embeddings.
> * **BrainTalker** \[BHI 2023]: A recent brain-to-speech decoding model built on Wav2Vec.
> * **CNN-LSTM** \[Nature 2023]: A widely used architecture in EEG decoding tasks. This model serves as one of our baselines, and its implementation is available on GitHub (linked in the appendix).
>
> The GitHub linked in the appendix already **included** the preprocessing scripts for extracting low- and high-frequency bands **at the time of submission**. We will release a cleaner HyperSpeech codebase and a public demo website with ground-truth vs. reconstructed audio upon acceptance.
>
> ---
>
> ### **Q9: Motivation for Euclidean Distance to Define Hyperedges.**
>    While Transformer models rely on dense, all-to-all attention, such connectivity can amplify neural noise—an inherent challenge in sEEG. In contrast, our **hypergraph-based approach introduces sparse and structured interactions**, grouping only functionally relevant channel subsets.
>
>    sEEG signals exhibit **localized, high-order interactions**, rather than global connectivity. Our hypergraph architecture efficiently captures this structure and **outperforms FastSpeech2** with only about **one-third the parameters** (6.6M vs. 18.2M), underscoring the strength of structured over dense attention.
> Notably, HyperSpeech introduces just **2.5% more parameters** (≈165K) than the CNN-LSTM baseline (6.44M vs. 6.61M), while incorporating a powerful spatiotemporal inductive bias through hypergraph modeling.
>
> ---
>
> ### **Q10: 2-Second Window.**
> The 2-second window (corresponding to 2000 data points) matches the utterance duration.
>
> ---
>
> ### **Q11: On BCI vs. Perceptual Neuroscience Goals.**
> While our current dataset focuses on stimulus-aligned brain responses, it lays the groundwork for future extensions involving perceptual reports. We see this release as a first step toward richer studies of auditory encoding.

---

> > ### Comment · Reviewer_L85g · 2025-08-02
> >
> > Thank you for your thorough replies. I am happy with the responses and think the work will be much easier to engage with after the promised changes. I appreciate the substantial effort.

---

> > > ### Author Response · Authors · 2025-08-02
> > > **Sincere Thanks to Reviewer L85g**
> > >
> > > Thank you sincerely for taking the time to read and respond. Your recognition means a great deal to us and greatly encourages our work. We truly appreciate your support!

---

### Official Review · Reviewer_AcXz · 2025-07-03

**Rating:** 5
**Confidence:** 3

**Summary:**

The authors presents a novel stereo-electroencephalography (sEEG) dataset for auditory speech reconstruction, and proposes HyperSpeech, a multi-band neural decoding framework using dynamic spatio-temporal hypergraph neural networks. The dataset is focused on densely sampled sEEG dataset from 5 clinical participants. The dataset is preprocessed and structured to be readily usable for machine learning applications.

**Dataset Code Accessibility:**

Partly

**Dataset Code Comments:**

Code for NeuroListen is provided, but not for HyperSpeech.

**Ethical Considerations:**

No, there are no or only very minor ethics concerns

**Final Justification:**

The authors acknowledged my concerns regarding missing information and committed to providing it. The dataset they present holds significant potential for advancing insights into BCI models. They have also addressed the other reviewers’ comments with care. Overall, both the dataset and the proposed method represent meaningful contributions to the field.

**Limitations Weaknesses:**

The use of the .npy format represents a notable limitation that is not explicitly addressed in the paper. This choice may hinder direct integration into lab-specific workflows or complicate interoperability. Furthermore, not sharing the raw data in its originally collected form (e.g., as EDF or similar clinical recording formats) restricts researchers from reprocessing the signals using their own pipelines, which is especially relevant for validating preprocessing steps, testing alternative referencing schemes, or applying custom artifact rejection methods.

Additionally, while the algorithm for HyperSpeech is described, the absence of accompanying code raises concerns regarding reproducibility.

**Strengths Contributions:**

Significance:
The authors collect and publicly release a clinical sEEG dataset. The dataset has the potential to provide new insights into BCI models and enable development of assistive communication technologies.

Quality:
The data collection methodology is well defined, with the appendix providing detailed supplementary information. Preprocessing and annotation procedures are clearly described and appropriately justified. The proposed method HyperSpeech consistently beats competitive baseline methods. The baseline evaluations are well scoped, and the chosen metrics are both appropriate and well motivated. Implementation details are thoroughly documented, and the code repository is publicly available. Additionally, the authors provide a thoughtful discussion of the dataset’s limitations and clearly articulate its potential value to the research community.

Clarity:
The paper is well written, easy to follow, and includes all the essential components expected in a data release manuscript. Key details are included in the supplementary material to complement the main text. Figures are well designed and aid in understanding. The motivation for dataset collection is clearly conveyed. Both data and anonymized code are accessible, and the ethics statement is complete and thoughtfully presented.

---

> ### Author Rebuttal · Authors · 2025-07-30
>
> ### **Q1: Flexible Format Support Upon Request.**
>
> We appreciate the reviewer’s feedback. While `.npy` was chosen for its **simplicity in Python-based workflows**, we acknowledge its limitations in interoperability.
> To address this, we plan to release the **original raw sEEG data** in **standard formats** (e.g., `.EEG` & `.EDF`), along with **preprocessing scripts** to enable **full reproducibility** and **custom reprocessing**.
> Furthermore, if the you have specific data needs or preferences, we would be happy to accommodate them to the extent possible.
> Our goal is to ensure that the dataset is **maximally usable** for future researchers, and we welcome feedback that would facilitate that.
> **Thank you for highlighting this important point.**
>
> ---
>
> ### **Q2: Willingness to Release Raw Data.**
>
> Thank you for your insightful comment.
> Because the preprocessing pipeline involves multiple **intricate steps**—including **mapping sEEG signals to corresponding audio segments**, **rejecting noisy or malfunctioning electrodes**, and **applying frequency‑domain filtering**—we have publicly released the **preprocessed dataset** to **facilitate usability**.
> We fully agree on the importance of **transparency and flexibility**; therefore, we will also release the **original raw data** together with the **complete preprocessing code** so that users can **inspect**, **reproduce**, or **customize** the workflow for their **specific research needs**.
>
> ---
>
> ### **Q3: HyperSpeech Availability.**
>
> We appreciate the concern.
> We have currently released the **baseline code necessary to reproduce the main results** in the paper.
> We will release the **HyperSpeech codebase** upon **paper acceptance**.

---

> > ### Comment · Reviewer_AcXz · 2025-08-04
> >
> > The authors acknowledged my concerns regarding data format and code availability, committed to providing it. The dataset they present holds significant potential for advancing insights into BCI models. They have also addressed the other reviewers’ comments with care. Overall, both the dataset and the proposed method represent meaningful contributions to the field.

---

### Decision · Program_Chairs · 2025-09-18

**Decision:**

Accept (poster)

**Comment:**

This paper introduces a stereo-electroencephalography (sEEG) dataset for auditory speech reconstruction and proposes HyperSpeech, a multi-band neural decoding framework leveraging dynamic spatio-temporal hypergraph neural networks. The authors demonstrate significance by publicly releasing a clinical sEEG dataset that can advance brain-computer interface (BCI) research and assistive communication technologies. The methodology is robust, with well-documented preprocessing and annotation processes, and the proposed HyperSpeech method consistently outperforms competitive baselines. The paper is clearly written, with comprehensive supplementary materials and accessible datasets, enhancing its value to the research community. However, the use of the .npy format, without providing raw data in clinical formats like EDF, limits interoperability and customization. Moreover, the absence of accompanying HyperSpeech code raises concerns about reproducibility.

===== FINAL UPDATE FROM DB Track PCs ====

The final decision for this paper has been taken by the program chairs after consultation with the SACs. All Senior Area Chairs have ranked papers according to the feedback from the AC during the review process. We decided to leave the original meta-review to reflect the opinion of the AC in light of the initial discussions with reviewers and SAC.